# Agnostic Estimation for Misspecified Phase Retrieval Models

**Matey Neykov**        **Zhaoran Wang**        **Han Liu**
Department of Operations Research and Financial Engineering
Princeton University, Princeton, NJ 08544
{mneykov, zhaoran, hanliu}@princeton.edu

## Abstract

The goal of noisy high-dimensional phase retrieval is to estimate an $s$-sparse parameter $\boldsymbol{\beta}^* \in \mathbb{R}^d$ from $n$ realizations of the model $Y = (\boldsymbol{X}^\top \boldsymbol{\beta}^*)^2 + \varepsilon$. Based on this model, we propose a significant semi-parametric generalization called *misspecified phase retrieval* (MPR), in which $Y = f(\boldsymbol{X}^\top \boldsymbol{\beta}^*, \varepsilon)$ with unknown $f$ and $\mathrm{Cov}(Y, (\boldsymbol{X}^\top \boldsymbol{\beta}^*)^2) > 0$. For example, MPR encompasses $Y = h(|\boldsymbol{X}^\top \boldsymbol{\beta}^*|) + \varepsilon$ with increasing $h$ as a special case. Despite the generality of the MPR model, it eludes the reach of most existing semi-parametric estimators. In this paper, we propose an estimation procedure, which consists of solving a cascade of two convex programs and provably recovers the direction of $\boldsymbol{\beta}^*$. Our theory is backed up by thorough numerical results.

## 1   Introduction

In scientific and engineering fields researchers often times face the problem of quantifying the relationship between a given outcome $Y$ and corresponding predictor vector $\boldsymbol{X}$, based on a sample $\{(Y_i, \boldsymbol{X}_i^\top)^\top\}_{i=1}^n$ of $n$ observations. In such situations it is common to postulate a linear "working" model, and search for a $d$-dimensional signal vector $\boldsymbol{\beta}^*$ satisfying the following familiar relationship:

$$Y = \boldsymbol{X}^\top \boldsymbol{\beta}^* + \varepsilon. \tag{1.1}$$

When the predictor $\boldsymbol{X}$ is high-dimensional in the sense that $d \gg n$, it is commonly assumed that the underlying signal $\boldsymbol{\beta}^*$ is $s$-sparse. In a certain line of applications, such as X-ray crystallography, microscopy, diffraction and array imaging[1], one can only measure the magnitude of $\boldsymbol{X}^\top \boldsymbol{\beta}^*$ but not its phase (i.e., sign in the real domain). In this case, assuming model (1.1) may not be appropriate. To cope with such applications in the high-dimensional setting, [7] proposed the thresholded Wirtinger flow (TWF), a procedure which consistently estimates the signal $\boldsymbol{\beta}^*$ in the following real sparse noisy phase retrieval model:

$$Y = (\boldsymbol{X}^\top \boldsymbol{\beta}^*)^2 + \varepsilon, \tag{1.2}$$

where one additionally knows that the predictors have a Gaussian random design $\boldsymbol{X} \sim \mathcal{N}(0, \mathbf{I}_d)$. In the present paper, taking an agnostic point of view, we recognize that both models (1.1) and (1.2) represent an idealized view of the data generating mechanism. In reality, the nature of the data could be better reflected through the more flexible viewpoint of a single index model (SIM):

$$Y = f(\boldsymbol{X}^\top \boldsymbol{\beta}^*, \varepsilon), \tag{1.3}$$

where $f$ is an unknown link function, and it is assumed that $\|\boldsymbol{\beta}^*\|_2 = 1$ for identifiability. A recent line of work on high-dimensional SIMs [25, 27], showed that under Gaussian designs, one can apply $\ell_1$ regularized least squares to successfully estimate the direction of $\boldsymbol{\beta}^*$ and its support. The crucial condition allowing for the above somewhat surprising application turns out to be:

$$\mathrm{Cov}(Y, \boldsymbol{X}^\top \boldsymbol{\beta}^*) \neq 0. \tag{1.4}$$

While condition (1.4) is fairly generic, encompassing cases with a binary outcome, such as logistic regression and one-bit compressive sensing [5], it fails to capture the phase retrieval model (1.2).

More generally, it is easy to see that when the link function $f$ is even in its first coordinate, condition (1.4) fails to hold. The goal of the present manuscript is to formalize a class of SIMs, which includes the noisy phase retrieval model as a special case in addition to various other additive and non-additive models with even link functions, and develop a procedure that can successfully estimate the direction of $\boldsymbol{\beta}^*$ up to a global sign. Formally, we consider models (1.3) with Gaussian design that satisfy the following moment assumption:

$$\mathrm{Cov}(Y, (\boldsymbol{X}^\top \boldsymbol{\beta}^*)^2) > 0. \tag{1.5}$$

Unlike (1.4), one can immediately check that condition (1.5) is satisfied by model (1.2). In §2 we give multiple examples, both abstract and concrete, of SIMs obeying this constraint. Our second moment constraint (1.5) can be interpreted as a semi-parametric robust version of phase-retrieval. Hence, we will refer to the class of models satisfying condition (1.5) as *misspecified phase retrieval* (MPR) models. In this point of view it is worth noting that condition (1.4) relates to linear regression in a way similar to how condition (1.5) relates to the phase retrieval model. Our motivation for studying SIMs under such a constraint can ultimately be traced to the vast sufficient dimension reduction (SDR) literature. In particular, we would like to point out [22] as a source of inspiration.

**Contributions.** Our first contribution is to formulate a novel and easily implementable two-step procedure, which consistently estimates the direction of $\boldsymbol{\beta}^*$ in an MPR model. In the first step we solve a semidefinite program producing a unit vector $\widehat{\mathbf{v}}$, such that $|\widehat{\mathbf{v}}^\top \boldsymbol{\beta}^*|$ is sufficiently large. Once such a pilot estimate is available, we consider solving an $\ell_1$ regularized least squares on the augmented outcome $\widetilde{Y}_i = (Y_i - \overline{Y})\boldsymbol{X}_i^\top \widehat{\mathbf{v}}$, where $\overline{Y}$ is the average of $Y_i$'s, to produce a second estimate $\widehat{\mathbf{b}}$, which is then normalized to obtain the final refined estimator $\widehat{\boldsymbol{\beta}} = \widehat{\mathbf{b}}/\|\widehat{\mathbf{b}}\|_2$. In addition to being universally applicable to MPR models, our procedure has an algorithmic advantage in that it relies solely on convex optimization, and as a consequence we can obtain the corresponding global minima of the two convex programs in polynomial time.

Our second contribution is to rigorously demonstrate that the above procedure consistently estimates the direction of $\boldsymbol{\beta}^*$. We prove that for a given MPR model, with high probability, one has:

$$\min_{\eta \in \{-1,1\}} \|\widehat{\boldsymbol{\beta}} - \eta \boldsymbol{\beta}^*\|_2 \lesssim \sqrt{s \log d/n},$$

provided that the sample size $n$ satisfies $n \gtrsim s^2 \log d$. While the same rates (with different constants) hold for the TWF algorithm of [7] in the special case of noisy phase retrieval model, our procedure provably achieves these rates over the broader class of MPR models.

Lastly, we propose an optional refinement of our algorithm, which shows improved performance in the numerical studies.

**Related Work.** The phase retrieval model has received considerable attention in the recent years by statistics, applied mathematics as well as signal processing communities. For the non-sparse version of (1.2), efficient algorithms have been suggested based on both semidefinite programs [8, 10] and non-convex optimization methods that extend gradient descent [9]. Additionally, a non-traditional instance of phase retrieval model (which also happens to be a special case of the MPR model) was considered by [11], where the authors suggested an estimation procedure originally proposed for the problem of mixed regression. For the noisy sparse version of model (1.2), near optimal solutions were achieved with a computationally infeasible program by [20]. Subsequently, a tractable gradient descent approach achieving minimax optimal rates was developed by [7].

Abstracting away from the phase retrieval or linear model settings, we note that inference for SIMs in the case when $d$ is small or fixed, has been studied extensively in the literature [e.g., 18, 24, 26, 34, among many others]. In another line of research on SDR, seminal insights shedding light on condition (1.4) can be found in, e.g., [12, 21, 23]. The modified condition (1.5) traces roots to [22], where the authors designed a procedure to handle precisely situations where (1.4) fails to hold. More recently, there have been active developments for high-dimensional SIMs. [27] and later [31] demonstrated that under condition (1.4), running the least squares with $\ell_1$ regularization can obtain a consistent estimate of the direction of $\boldsymbol{\beta}^*$, while [25] showed that this procedure also recovers the signed support of the direction. Excess risk bounds were derived in [14]. Very recently, [16] extended this observation to other convex loss functions under a condition corresponding to (1.4) depending implicitly on the loss function of interest. [28] proposed a non-parametric least squares with an equality $\ell_1$ constraint to handle simultaneous estimation of $\boldsymbol{\beta}^*$ as well as $f$. [17] considered a smoothed-out $U$-process type of loss function with $\ell_1$ regularization, and proved their approach works for a sub-class of functions satisfying condition (1.4). None of the aforementioned works on SIMs can be directly applied to tackle the MPR class (1.5). A generic procedure for estimating sparse principal eigenvectors was

developed in [37]. While in principle this procedure can be applied to estimate the direction in MPR models, it requires proper initialization, and in addition, it requires knowledge of the sparsity of the vector $\boldsymbol{\beta}^*$. We discuss this approach in more detail in §4.

Regularized procedures have also been proposed for specific choices of $f$ and $Y$. For example, [36] studied consistent estimation under the model $\mathbb{P}(Y = 1|\boldsymbol{X}) = (h(\boldsymbol{X}^\top \boldsymbol{\beta}^*) + 1)/2$ with binary $Y$, where $h : \mathbb{R} \mapsto [-1, 1]$ is possibly unknown. Their procedure is based on taking pairs of differences in the outcome, and therefore replaces condition (1.4) with a different type of moment conditon. [35] considered the model $Y = h(\boldsymbol{X}^\top \boldsymbol{\beta}^*) + \varepsilon$ with a known continuously differentiable and monotonic $h$, and developed estimation and inferential procedures based on the $\ell_1$ regularized quadratic loss, in a similar spirit to the TWF algorithm suggested by [7]. In conclusion, although there exists much prior related work, to the best of our knowledge, none of the available literature discusses the MPR models in the generality we attempt in the present manuscript.

**Notation.** We now briefly outline some commonly used notations. Other notations will be defined as needed throughout the paper. For a (sparse) vector $\mathbf{v} = (v_1, \ldots, v_p)^\top$, we let $S_{\mathbf{v}} := \text{supp}(\mathbf{v}) = \{j : v_j \neq 0\}$ denote its support, $\|\mathbf{v}\|_p$ denote the $\ell_p$ norm (with the usual extension when $p = \infty$) and $\mathbf{v}^{\otimes 2} := \mathbf{v}\mathbf{v}^\top$ is a shorthand for the outer product. With a standard abuse of notation we will denote by $\|\mathbf{v}\|_0 = |\text{supp}(\mathbf{v})|$ the cardinality of the support of $\mathbf{v}$. We often use $\mathbf{I}_d$ to denote a $d \times d$ identity matrix. For a real random variable $X$, define

$$\|X\|_{\psi_2} = \sup_{p \geq 1} p^{-1/2} (\mathbb{E}|X|^p)^{1/p}, \quad \|X\|_{\psi_1} = \sup_{p \geq 1} p^{-1} (\mathbb{E}|X|^p)^{1/p}.$$

Recall that a random variable is called *sub-Gaussian* if $\|X\|_{\psi_2} < \infty$ and *sub-exponential* if $\|X\|_{\psi_1} < \infty$ [e.g., 32]. For any integer $k \in \mathbb{N}$ we use the shorthand notation $[k] = \{1, \ldots, k\}$. We also use standard asymptotic notations. Given two sequences $\{a_n\}, \{b_n\}$ we write $a_n = O(b_n)$ if there exists a constant $C < \infty$ such that $a_n \leq Cb_n$, and $a_n \asymp b_n$ if there exist positive constants $c$ and $C$ such that $c < a_n/b_n < C$.

**Organization.** In §2 and §3 we introduce the MPR model class and our estimation procedure, and §3.1 is dedicated to stating the theoretical guarantees of our proposed algorithm. Simulation results are given in §4. A brief discussion is provided in §5. We defer the proofs to the appendices due to space limitations.

## 2 MPR Models

In this section we formally introduce MPR models. In detail, we argue that the class of such models is sufficiently rich, including numerous models of interest. Motivated by the setup in the sparse noisy phase retrieval model (1.2), we assume throughout the remainder of the paper that $\boldsymbol{X} \sim \mathcal{N}(0, \mathbf{I}_d)$. We begin our discussion with a formal definition.

**Definition 2.1** (MPR Models). Assume that we are given model (1.3), where $\boldsymbol{X} \sim \mathcal{N}(0, \mathbf{I}_d), \varepsilon \perp\!\!\!\perp \boldsymbol{X}$ and $\boldsymbol{\beta}^* \in \mathbb{R}^d$ is an $s$-sparse unit vector, i.e., $\|\boldsymbol{\beta}^*\|_2 = 1$. We call such a model *misspecified phase retrieval* (MPR) model, if the link function $f$ and noise $\varepsilon$ further satisfy, for $Z \sim \mathcal{N}(0, 1)$ and $K > 0$,

$$c_0 := \text{Cov}(f(Z, \varepsilon), Z^2) > 0, \qquad (2.1) \qquad \qquad \|Y\|_{\psi_1} \leq K. \qquad (2.2)$$

Both assumptions (2.1) and (2.2) impose moment restrictions on the random variable $Y$. Assumption (2.1) states that $Y$ is positively correlated with the random variable $(\boldsymbol{X}^\top \boldsymbol{\beta}^*)^2$, while assumption (2.2) requires $Y$ to have somewhat light-tails. Also, as mentioned in the introduction, the unit norm constraint on the vector $\boldsymbol{\beta}^*$ is required for the identifiability of model (1.3). We remark that the class of MPR models is convex in the sense that if we have two MPR models $f_1(\boldsymbol{X}^\top \boldsymbol{\beta}^*, \varepsilon)$ and $f_2(\boldsymbol{X}^\top \boldsymbol{\beta}^*, \varepsilon)$, all models generated by their convex combinations $\lambda f_1(\boldsymbol{X}^\top \boldsymbol{\beta}^*, \varepsilon) + (1-\lambda) f_2(\boldsymbol{X}^\top \boldsymbol{\beta}^*, \varepsilon)$ $(\lambda \in [0, 1])$ are also MPR models. It is worth noting the $>$ direction in (2.1) is assumed without loss of generality. If $\text{Cov}(Y, (\boldsymbol{X}^\top \boldsymbol{\beta}^*)^2) < 0$ one can apply the same algorithm to $-Y$. However, the knowledge of the direction of the inequality is important.

In the following, we restate condition (2.1) in a more convenient way, enabling us to easily calculate the explicit value of the constant $c_0$ in several examples.

**Proposition 2.2.** Assume that there exists a version of the map $\varphi(z) : z \mapsto \mathbb{E}[f(Z, \varepsilon)|Z = z]$ such that $\mathbb{E}D^2\varphi(Z) > 0$, where $D^2$ is the second distributional derivative of $\varphi$ and $Z \sim \mathcal{N}(0, 1)$. Then the SIM (1.3) satisfies assumption (2.1) with $c_0 = \mathbb{E}D^2\varphi(Z)$.

We now provide three concrete MPR models as warm up examples for the more general examples discussed in Proposition 2.3 and Remark 2.3. Consider the models:

$$Y = (\boldsymbol{X}^\top \boldsymbol{\beta}^*)^2 + \varepsilon, \quad (2.3) \qquad Y = |\boldsymbol{X}^\top \boldsymbol{\beta}^*| + \varepsilon, \quad (2.4) \qquad Y = |\boldsymbol{X}^\top \boldsymbol{\beta}^* + \varepsilon|, \quad (2.5)$$

where $\varepsilon \perp\!\!\!\perp \boldsymbol{X}$ is sub-exponential noise, i.e., $\|\varepsilon\|_{\psi_1} \leq K_\varepsilon$ for some $K_\varepsilon > 0$. Model (2.3) is the noisy phase retrieval model considered by [7], while models (2.4) and (2.5) were both discussed in [11], where the authors proposed a method to solve model (2.5) in the low-dimensional regime. Below we briefly explain why these models satisfy conditions (2.1) and (2.2). First, observe that in all three models we have a sum of two sub-exponential random variables, and hence by the triangle inequality it follows that the random variable $Y$ is also sub-exponential, which implies (2.2). To understand why (2.1) holds, by applying Proposition 2.2 we have $c_0 = \mathbb{E}2 = 2 > 0$ for model (2.3), $c_0 = \mathbb{E}2\delta_0(Z) = 2/\sqrt{2\pi} > 0$ for model (2.4), and $c_0 = \mathbb{E}2\delta_0(Z + \varepsilon) = 2\mathbb{E}\phi(\varepsilon) > 0$ for model (2.5), where $\delta_0(\cdot)$ is the Dirac delta function centered at zero, and $\phi$ is the density of the standard normal distribution.

Admittedly, calculating the second distributional derivative could be a laborious task in general. In the remainder of this section we set out to find a simple to check generic sufficient condition on the link function $f$ and error term $\varepsilon$, under which both (2.1) and (2.2) hold. Before giving our result note that the condition $\mathbb{E}D^2\varphi(Z) > 0$ is implied whenever $\varphi$ is strictly convex and twice differentiable. However, strictly convex functions $\varphi$ may violate assumption (2.2) as they can inflate the tails of $Y$ arbitrarily (consider, e.g., $f(x, \varepsilon) = x^4 + \varepsilon$). Moreover, the functions in example (2.4) and (2.5) fail to be twice differentiable. In the following result we handle those two problems, and in addition we provide a more generic condition than convexity, which suffices to ensure the validity of (2.1).

**Proposition 2.3.** The following statements hold.
  (i) Let the function $\varphi$ defined in Proposition 2.2 be such that the map $z \mapsto \varphi(z) + \varphi(-z)$ is non-decreasing on $\mathbb{R}_0^+$ and and there exist $z_1 > z_2 > 0$ such that $\varphi(z_1) + \varphi(-z_1) > \varphi(z_2) + \varphi(-z_2)$. Then (2.1) holds.
  (ii) A sufficient condition for (i) to hold, is that $z \mapsto \varphi(z)$ is convex and sub-differentiable at every point $z \in \mathbb{R}$, and there exists a point $z_0 \in \mathbb{R}_0^+$ satisfying $\varphi(z_0) + \varphi(-z_0) > 2\varphi(0)$.
  (iii) Assume that there exist functions $g_1, g_2$ such that $f(Z, \varepsilon) \leq g_1(Z) + g_2(\varepsilon)$, and $g_1$ is *essentially quadratic* in the sense that there exists a closed interval $\mathcal{I} = [a, b]$ with $0 \in \mathcal{I}$, such that for all $z$ satisfying $g_1(z) \in \mathcal{I}^c$ we have $|g_1(z)| \leq Cz^2$ for a sufficiently large constant $C > 0$, and let $g_2(\varepsilon)$ be sub-exponential. Then (2.2) holds.

**Remark 2.4.** Proposition 2.3 shows that the class of MPR models is sufficiently broad. By (i) and (ii) it immediately follows that the additive models

$$Y = h(\boldsymbol{X}^\top \boldsymbol{\beta}^*) + \varepsilon, \tag{2.6}$$

where the link function $h$ is even and increasing on $\mathbb{R}_0^+$ or convex, satisfy the covariance condition (2.1) by (i) and (ii) of Proposition 2.3 respectively. If $h$ is also essentially quadratic and $\varepsilon$ is sub-exponentially distributed, using (iii) we can deduce that $Y$ in (2.6) is a sub-exponential random variable, and hence under these restrictions model (2.6) is an MPR model. Both examples (2.3) and (2.4) take this form.

Additionally, Proposition 2.3 implies that the model

$$Y = h(\boldsymbol{X}^\top \boldsymbol{\beta}^* + \varepsilon) \tag{2.7}$$

satisfies (2.1), whenever the link $h$ is a convex sub-differentiable function, such that $h(z_0) + h(-z_0) > 2h(0)$ for some $z_0 > 0$, $\mathbb{E}|h(z + \varepsilon)| < \infty$ for all $z \in \mathbb{R}$ and $\mathbb{E}|h(Z + \varepsilon)| < \infty$. This conclusion follows because under the latter conditions the function $\varphi(z) = \mathbb{E}h(z + \varepsilon)$ satisfies (ii), which is proved in Appendix C under Lemma C.1. Moreover, if it turns out that $h$ is essentially quadratic and $h(2\varepsilon)$ is sub-exponential, then by Jensen's inequality we have $2h(Z + \varepsilon) \leq h(2Z) + h(2\varepsilon)$ and hence (iii) implies that (2.2) is also satisfied. Model (2.5) is of the type (2.7). Unlike the additive noise models (2.6), models (2.7) allow noise corruption even within the argument of the link function. On the negative side, it should be apparent that (2.1) fails to hold in cases where $\varphi$ is an odd function, i.e., $\varphi(z) = -\varphi(-z)$. In many such cases (e.g. when $\varphi$ is monotone or non-constant and non-positive/non-negative on $\mathbb{R}^+$), one would have $\mathrm{Cov}(Y, \boldsymbol{X}^\top \boldsymbol{\beta}^*) = \mathbb{E}[\varphi(Z)Z] \neq 0$, and hence direct application of the $\ell_1$ regularized least squares algorithm is possible as we discussed in the introduction.

# 3 Agnostic Estimation for MPR

In this section we describe and motivate our two-step procedure, which consists of a convex relaxation and an $\ell_1$ regularized least squares program, for performing estimation in the MPR class of models

described by Definition 2.1. We begin our motivation by noting that any MPR model satisfies the following inequality

$$\mathrm{Cov}((Y - \mu)\boldsymbol{X}^\top\boldsymbol{\beta}^*, \boldsymbol{X}^\top\boldsymbol{\beta}^*) = \mathbb{E}\{(Y - \mu)(\boldsymbol{X}^\top\boldsymbol{\beta}^*)^2\} = \mathrm{Cov}(f(Z, \varepsilon), Z^2) = c_0 > 0, \quad (3.1)$$

where we have denoted $\mu := \mathbb{E}Y$. This simple observation plays a major role in the motivation of our procedure. Notice that in view of condition (1.4), inequality (3.1) implies that if instead of observing $Y$ we had observed $\breve{Y} = g(\boldsymbol{X}^\top\boldsymbol{\beta}^*, \varepsilon) = (Y - \mu)\boldsymbol{X}^\top\boldsymbol{\beta}^*$. However, there is no direct way of generating the random variable $\breve{Y}$, as doing so would require the knowledge of $\boldsymbol{\beta}^*$ and the mean $\mu$. Here, we propose to roughly estimate $\boldsymbol{\beta}^*$ by a vector $\widehat{\mathbf{v}}$ first, use an empirical estimate $\overline{Y}$ of $\mu$, and then obtain the $\ell_1$ regularized least squares estimate on the augmented variable $\widetilde{Y} = (Y - \overline{Y})\boldsymbol{X}^\top\widehat{\mathbf{v}}$ to sharpen the convergence rate. At first glance it might appear counter-intuitive that introducing a noisy estimate of $\boldsymbol{\beta}^*$ can lead to consistent estimates, as the so-defined $\widetilde{Y}$ variable depends on the projection of $\boldsymbol{X}$ on $\mathrm{span}\{\boldsymbol{\beta}^*, \widehat{\mathbf{v}}\}$. Decompose

$$\widehat{\mathbf{v}} = (\widehat{\mathbf{v}}^\top\boldsymbol{\beta}^*)\boldsymbol{\beta}^* + \widehat{\boldsymbol{\beta}}^\perp, \quad (3.2)$$

where $\widehat{\boldsymbol{\beta}}^\perp \perp \boldsymbol{\beta}^*$. To better motivate this proposal, in the following we analyze the population least squares fit, based on the augmented variable $\breve{Y} = (Y - \mu)\boldsymbol{X}^\top\widehat{\mathbf{v}}$ for some *fixed* unit vector $\widehat{\mathbf{v}}$ with decomposition (3.2). Writing out the population solution for least squares yields:

$$[\mathbb{E}\boldsymbol{X}^{\otimes 2}]^{-1}\mathbb{E}[\boldsymbol{X}\breve{Y}] = \underbrace{\mathbb{E}[\boldsymbol{X}(Y - \mu)\boldsymbol{X}^\top(\widehat{\mathbf{v}}^\top\boldsymbol{\beta}^*)\boldsymbol{\beta}^*]}_{\boldsymbol{I}_1} + \underbrace{\mathbb{E}[\boldsymbol{X}(Y - \mu)\boldsymbol{X}^\top\widehat{\boldsymbol{\beta}}^\perp]}_{\boldsymbol{I}_2}. \quad (3.3)$$

We will now argue that left hand side of (3.3) is proportional to $\boldsymbol{\beta}^*$. First, we observe that $\boldsymbol{I}_1 = c_0(\widehat{\mathbf{v}}^\top\boldsymbol{\beta}^*)\boldsymbol{\beta}^*$, since multiplying by any vector $\mathbf{b} \perp \boldsymbol{\beta}^*$ yields $\mathbf{b}^\top\boldsymbol{I}_1 = 0$ by independence. Second, and perhaps more importantly, we have that $\boldsymbol{I}_2 = 0$. To see this, we first take a vector $\mathbf{b} \in \mathrm{span}\{\boldsymbol{\beta}^*, \widehat{\boldsymbol{\beta}}^\perp\}^\perp$. Since the three variables $\mathbf{b}^\top\boldsymbol{X}, Y - \mu$ and $\widehat{\boldsymbol{\beta}}^\perp\boldsymbol{X}$ are independent, we have $\mathbf{b}^\top\boldsymbol{I}_2 = 0$. Multiplying by $\boldsymbol{\beta}^*$ we have $\boldsymbol{\beta}^{*\top}\boldsymbol{I}_2 = 0$ since $\boldsymbol{\beta}^{*\top}\boldsymbol{X}(Y - \mu)$ is independent of $\boldsymbol{X}^\top\widehat{\boldsymbol{\beta}}^\perp$. Finally, multiplying by $\widehat{\boldsymbol{\beta}}^\perp$ yields $\boldsymbol{I}_2^\top\widehat{\boldsymbol{\beta}}^\perp = 0$, since $(\boldsymbol{X}^\top\widehat{\boldsymbol{\beta}}^\perp)^2$ is independent of $Y - \mu$.

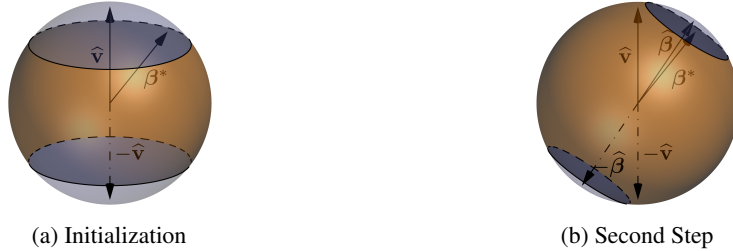

(a) Initialization        (b) Second Step

Figure 1: An illustration of the estimates $\widehat{\mathbf{v}}$ and $\widehat{\boldsymbol{\beta}}$ produced by the first and second steps of Algorithm 1. After the first step we can guarantee that the vector $\boldsymbol{\beta}^*$ belongs to one of two spherical caps which contain all vectors $\mathbf{w}$ such that $|\widehat{\mathbf{v}}^\top\mathbf{w}| \geq \kappa$ for some constant $\kappa > 0$, provided that the sample size $n \gtrsim s^2 \log d$ is sufficiently large. After the second step we can guarantee that the vector $\boldsymbol{\beta}^*$ belongs to one of two spherical caps in (b), which are shrinking with $(n, s, d)$ at a faster rate.

It is noteworthy to mention that the above derivation crucially relies on the fact that the $Y$ variable was centered, and the vector $\widehat{\mathbf{v}}$ was fixed. In what follows we formulate a pilot procedure which produces an estimate $\widehat{\mathbf{v}}$ such that $|\widehat{\mathbf{v}}^\top\boldsymbol{\beta}^*| \geq \kappa > 0$. A proper initialization algorithm can be achieved by using a spectral method, such as the Principal Hessian Directions (PHD) proposed by [22]. Cast into the framework of SIM, the PHD framework implies the following simple observation:

**Lemma 3.1.** If we have an MPR model, then $\mathrm{argmax}_{\|\mathbf{v}\|_2=1} \mathbf{v}^\top\mathbb{E}[Y(\boldsymbol{X}^{\otimes 2} - \mathbf{I})]\mathbf{v} = \pm\boldsymbol{\beta}^*$.

A proof of this fact can be found in Appendix C. Lemma 3.1 encourages us to look into the following sample version maximization problem

$$\mathrm{argmax}_{\|\mathbf{v}\|_2=1, \|\mathbf{v}\|_0=s} n^{-1}\mathbf{v}^\top\textstyle\sum_{i=1}^n [Y_i(\boldsymbol{X}_i^{\otimes 2} - \mathbf{I})]\mathbf{v}, \quad (3.4)$$

which targets a restricted ($s$-sparse) principal eigenvector. Unfortunately, solving such a problem is a computationally intensive task, and requires knowledge of $s$. Here we take a standard route of relaxing the above problem to a convex program, and solving it efficiently via semidefinite programming (SDP). A similar in spirit SDP relaxation for solving sparse PCA problems, was originally proposed by [13]. Instead of solving (3.4), define $\widehat{\boldsymbol{\Sigma}} = n^{-1}\sum_{i=1}^n Y_i(\boldsymbol{X}_i^{\otimes 2} - \mathbf{I})$, and solve the following convex

program:

$$\widehat{\mathbf{A}} = \operatorname{argmax}_{\operatorname{tr}(\mathbf{A})=1, \mathbf{A} \in \mathbb{S}_+^d} \operatorname{tr}(\widehat{\mathbf{\Sigma}}\mathbf{A}) - \lambda_n \sum_{i,j=1}^d |A_{ij}|, \tag{3.5}$$

where $\mathbb{S}_+^d$ is the convex cone of non-negative semidefinite matrices, and $\lambda_n$ is a regularization parameter encouraging element-wise sparsity in the matrix $\mathbf{A}$. The hopes of introducing the optimization program above are that $\widehat{\mathbf{A}}$ will be a good first estimate of $\boldsymbol{\beta}^{*\otimes 2}$. In practice it could turn out that the matrix $\widehat{\mathbf{A}}$ is not rank one, hence we suggest taking $\widehat{\mathbf{v}}$ as the principal eigenvector of $\widehat{\mathbf{A}}$. In theory we show that with high probability the matrix $\widehat{\mathbf{A}}$ will indeed be of rank one. Observation (3.3), Lemma 3.1 and the SDP formulation motivate the agnostic two-step estimation procedure for misspecified phase retrieval in Algorithm 1.

---
**Algorithm 1**

---
**input** :$(Y_i, \boldsymbol{X}_i)_{i=1}^n$: data, $\lambda_n, \nu_n$: tuning parameters
1. Split the sample into two approximately equal sets $S_1$, $S_2$, with $|S_1| = \lfloor n/2 \rfloor$, $|S_2| = \lceil n/2 \rceil$.
2. Let $\widehat{\mathbf{\Sigma}} := |S_1|^{-1} \sum_{i \in S_1} Y_i (\boldsymbol{X}_i^{\otimes 2} - \mathbf{I}_d)$. Solve (3.5). Let $\widehat{\mathbf{v}}$ be the first eigenvector of $\widehat{\mathbf{A}}$.
3. Let $\overline{Y} = |S_2|^{-1} \sum_{i \in S_2} Y_i$. Solve the following program:

$$\widehat{\mathbf{b}} = \operatorname{argmin}_{\mathbf{b}} (2|S_2|)^{-1} \sum_{i \in S_2} ((Y_i - \overline{Y})\boldsymbol{X}_i^\top \widehat{\mathbf{v}} - \boldsymbol{X}_i^\top \mathbf{b})^2 + \nu_n \|\mathbf{b}\|_1. \tag{3.6}$$

4. Return $\widehat{\boldsymbol{\beta}} := \widehat{\mathbf{b}}/\|\widehat{\mathbf{b}}\|_2$.

---

The sample split is required to ensure that after decomposition (3.2), the vector $\widehat{\boldsymbol{\beta}}^\perp$ and the value $\widehat{\mathbf{v}}^\top \boldsymbol{\beta}^*$ are independent of the remaining sample. In §3.1 we demonstrate that Algorithm 1 succeeds with optimal (in the noisy regime) $\ell_2$ rate $\sqrt{s \log d/n}$, provided that $s^2 \log d \lesssim n$. The latter requirement on the sample size suffices to guarantee that the solution $\widehat{\mathbf{A}}$ of optimization program (3.5) is rank one. Figure 1 illustrates the two steps of Algorithm 1. In addition to our main procedure, we propose an optional refinement step (Algorithm 2) in which one applies steps 3. and 4. of Algorithm 1 on the full dataset using the output vector $\widehat{\boldsymbol{\beta}}$ of Algorithm 1. Doing so can potentially result in additional stability and further refinements of the rate constant.

---
**Algorithm 2** Optional Refinement

---
**input** :$(Y_i, \boldsymbol{X}_i)_{i=1}^n$: data, $\nu_n'$: tuning parameter, output $\widehat{\boldsymbol{\beta}}$ from the Algorithm 1
5. Let $\overline{Y} = n^{-1} \sum_{i \in [n]} Y_i$. Solve the following program:

$$\widehat{\mathbf{b}} = \operatorname{argmin}_{\mathbf{b}} (2n)^{-1} \sum_{i=1}^n ((Y_i - \overline{Y})\boldsymbol{X}_i^\top \widehat{\boldsymbol{\beta}} - \boldsymbol{X}_i^\top \mathbf{b})^2 + \nu_n' \|\mathbf{b}\|_1. \tag{3.7}$$

6. Return $\widehat{\boldsymbol{\beta}}' := \widehat{\mathbf{b}}/\|\widehat{\mathbf{b}}\|_2$.

---

### 3.1 Theoretical Guarantees

In this section we present our main theoretical results, which consist of theoretical justification of our procedures, as well as lower bounds for certain types of SIM (1.3). To simplify the presentation for this section, we slightly change the notation and assume that the sample size is $2n$ and $S_1 = [n]$ and $S_2 = \{n+1, \dots, 2n\}$. Of course this abuse of notation does not restrict our analysis to only even sample size cases.

Our first result shows that the optimization program (3.5) succeeds in producing a vector $\widehat{\mathbf{v}}$ which is close to the vector $\boldsymbol{\beta}^*$.

**Proposition 3.2.** Assume that $n$ is large enough so that $s\sqrt{\log d/n} < (1/6 - \kappa/4)c_0/(C_1 + C_2)$ for some small but fixed $\kappa > 0$ and constants $C_1, C_2$ (depending on $f$ and $\varepsilon$). Then there exists a value of $\lambda_n \asymp \sqrt{\log d/n}$ such that the principal eigenvector $\widehat{\mathbf{v}}$ of $\widehat{\mathbf{A}}$, the solution of (3.5), satisfies

$$|\widehat{\mathbf{v}}^\top \boldsymbol{\beta}^*| \geq \kappa > 0,$$

with probability at least $1 - 4d^{-1} - O(n^{-1})$.

Proposition 3.2 shows that the first step of Algorithm 1 narrows down the search for the direction of $\boldsymbol{\beta}^*$ to a union of two spherical caps (i.e., the estimate $\widehat{\mathbf{v}}$ satisfies $|\widehat{\mathbf{v}}^\top \boldsymbol{\beta}^*| \geq \kappa$ for some constant $\kappa > 0$, see also Figure 1a). Our main result below, demonstrates that in combination with program (3.6) this suffices to recover the direction of $\boldsymbol{\beta}^*$ at an optimal rate with high probability.

**Theorem 3.3.** There exist values of $\lambda_n, \nu_n \asymp \sqrt{\log d/n}$ and a constant $R > 0$ depending on $f$ and $\varepsilon$, such that if $s\sqrt{\log d/n} < R$ and $\log(d)\log^2(n)/n = o(1)$, the output of Algorithm 1 satisfies:

$$\sup_{\|\boldsymbol{\beta}^*\|_2=1, \|\boldsymbol{\beta}^*\|_0 \le s} \mathbb{P}_{\boldsymbol{\beta}^*}\left( \min_{\eta \in \{1,-1\}} \|\widehat{\boldsymbol{\beta}} - \eta\boldsymbol{\beta}^*\|_2 > L\sqrt{\frac{s \log d}{n}} \right) \le O(d^{-1} \vee n^{-1}), \qquad (3.8)$$

where $L$ is a constant depending solely on $f$ and $\varepsilon$.

We remark that although the estimation rate is of the order $\sqrt{s \log d/n}$, our procedure still requires that $s\sqrt{\log d/n}$ is sufficiently small. This phenomenon is similar to what has been observed by [7], and it is our belief that this requirement cannot be relaxed for computationally feasible algorithms. We would further like to mention that while in bound (3.8) we control the worst case probability of failure, it is less clear whether the estimate $\widehat{\boldsymbol{\beta}}$ is universally consistent (i.e., whether the sup can be moved inside the probability in (3.8)).

## 4 Numerical Experiments

In this section we provide numerical experiments based on the three models (2.3), (2.4) and (2.5) where the random variable $\varepsilon \sim \mathcal{N}(0, 1)$. All models are compared with the Truncated Power Method (TPM), proposed in [37]. For model (2.3) we also compare the results of our approach to the ones given by the TWF algorithm of [7]. Our setup is as follows. In all scenarios the vector $\boldsymbol{\beta}^*$ was held fixed at $\boldsymbol{\beta}^* = (\underbrace{-s^{-1/2}, s^{-1/2}, \dots, s^{-1/2}}_{s}, \underbrace{0, \dots 0}_{d-s})$. Since our theory requires that $n \gtrsim s^2 \log d$, we have setup four different sample sizes $n \approx \theta s^2 \log d$, where $\theta \in \{4, 8, 12, 16\}$. We let $s$ depend on the dimension $d$ and we take $s \approx \log d$. In addition to the suggested approach in Algorithm 1, we also provide results using the refinement procedure (see Algorithm 3.7). We also provide the values of two "warm" starts of our algorithm, produced by solving program (3.5) with half and full data correspondingly. It is evident that for all scenarios the second step of Algorithms 1 and 2 outperform the warm start from SDP, except in Figure 2 (b), (c), when the sample size is simply two small to for the warm start on half of the data to be accurate. All values we report are based on an average over 100 simulations.

The SDP parameter was kept at a constant value (0.015) throughout all simulations, and we observed that varying this parameter had little influence on the final SDP solution. To select the $\nu_n$ parameter for (3.6) a pre-specified grid of parameters $\{\nu^1, \dots, \nu^l\}$ was selected, and the following heuristic procedure based on $K$-fold cross-validation was used. We divide $S_2$ into $K = 5$ approximately equally sized non-intersecting sets $S_2 = \cup_{j \in [K]} \widetilde{S}_2^j$. For each $j \in [K]$ and $k \in [l]$ we run (3.6) on the set $\cup_{r \in [K], r \neq j} \widetilde{S}_2^r$ with a tuning parameter $\nu_n = \nu^k$ to obtain an estimate $\widehat{\boldsymbol{\beta}}_{k, -\widetilde{S}_2^j}$. Lemma 3.1 then justifies the following criteria to select the optimal index for selecting $\widehat{\nu}_n = \nu^{\widehat{l}}$ where

$$\widehat{l} = \operatorname*{argmax}_{k \in [l]} \sum_{j \in [K]} \sum_{i \in \widetilde{S}_2^j} Y_i (\boldsymbol{X}_i^\top \widehat{\boldsymbol{\beta}}_{k, -\widetilde{S}_2^j})^2.$$

Our experience suggests this approach works well in practice provided that the values $\{\nu^1, \dots, \nu^l\}$ are selected within appropriate range and are of the magnitude $\sqrt{\log d/n}$.

Since the TPM algorithm requires an estimate of the sparsity $s$, we tuned it as suggested in Section 4.1.2 of [37]. In particular, for each scenario we considered the set of possible sparsities $K = \{s, 2s, 4s, 8s\}$. For each $k \in K$ the algorithm is ran on the first part of the data $S_1$, to obtain an estimate $\widehat{\boldsymbol{\beta}}_k$, and the final estimate is taken to be $\widehat{\boldsymbol{\beta}}_{\widehat{k}}$ where $\widehat{k}$ is given by

$$\widehat{k} = \operatorname*{argmax}_{k \in K} \widehat{\boldsymbol{\beta}}_k^\top |S_2|^{-1} \sum_{i \in S_2} Y_i (\boldsymbol{X}_i^{\otimes 2} - \mathbf{I}_d)\widehat{\boldsymbol{\beta}}_k.$$

The TPM is ran for 2000 iterations. In the case of phase retrieval, the TWF algorithm was also ran at a total number of 2000 iterations, using the tuning parameters originally suggested in [7]. As expected the TWF algorithm which targets the sparse phase retrieval model in particular outperforms our approach in the case when the sample size $n$ is small, however our approach performs very comparatively to the TWF, and in fact even slightly better once we increase the sample size. It is possible that the TWF algorithm can perform better if it is ran for a longer than 2000 iterations, though in most cases it appeared to have converged to its final value. The results are visualized on Figure 2 above. The TPM algorithm, has performance comparable to that of Algorithm 1, is always

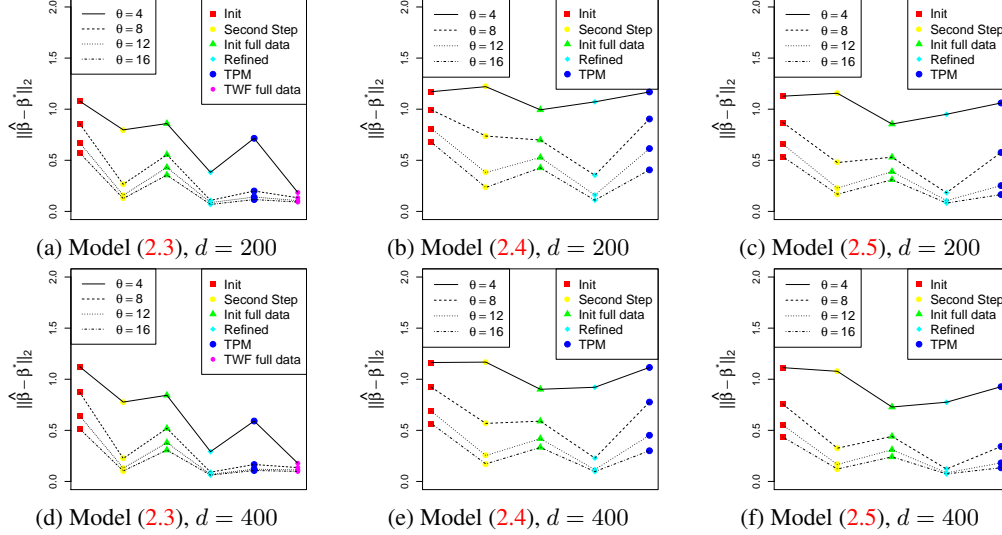

Figure 2: Simulation results for the three examples considered in §2, in two different settings for the dimension $d = 200, 400$. Here the parameter $\theta \approx \frac{n}{s^2 \log d}$ describes the relationship between sample size, dimension and sparsity of the signal. Algorithm 2 dominates in most settings, with exceptions when $\theta$ is too small, in which case none of the approaches provides meaningful results.

worse than the estimate produced by Algorithm 2, and it needs an initialization (for the first step of Algorithm 1 is used) and further requires a rough knowledge of the sparsity $s$, whereas both Algorithms 1 and 2 do not require an estimate of $s$.

## 5   Discussion

In this paper we proposed a two-step procedure for estimation of MPR models with standard Gaussian designs. We argued that the MPR models form a rich class including numerous additive SIMs (i.e., $Y = h(\boldsymbol{X}^\top \boldsymbol{\beta}^*) + \varepsilon$) with an even and increasing on $\mathbb{R}^+$ link function $h$. Our algorithm is based solely on convex optimization, and achieves optimal rates of estimation.

Our procedure does require that the sample size $n \gtrsim s^2 \log d$ to ensure successful initialization. The same condition has been exhibited previously, e.g., in [7] for the phase retrieval model, and in works on sparse principal components analysis [see, e.g., 3, 15, 33]. We anticipate that for a certain subclass of MPR models, the sample size requirement $n \gtrsim s^2 \log d$ is necessary for computationally efficient algorithms to exist. We conjecture that models (2.3)-(2.5) are such models. It is however certainly not true that this sample size requirement holds for all models from the MPR class. For example, the following model can be solved efficiently by applying the Lasso algorithm, without requiring the sample size scaling $n \gtrsim s^2 \log d$

$$Y = \text{sign}(\boldsymbol{X}^\top \boldsymbol{\beta}^* + c),$$

where $c < 0$ is fixed. This discussion leads to the important question under what conditions of the (known) link and error distribution $(f, \varepsilon)$ one can efficiently solve the SIM $Y = f(\boldsymbol{X}^\top \boldsymbol{\beta}^*, \varepsilon)$ with an optimal sample complexity. We would like to investigate this issue further in our future work.

**Acknowledgments:** The authors would like to thank the reviewers and meta-reviewers for carefully reading the manuscript and their helpful suggestions which improved the presentation. The authors would also like to thank Professor Xiaodong Li for kindly providing the code for the TWF algorithm.

## Footnotes

[1] In such applications it is typically assumed that $\boldsymbol{X} \in \mathbb{C}^d$ is a complex normal random vector. In this paper for simplicity we only consider the real case $\boldsymbol{X} \in \mathbb{R}^d$.

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
