[Supplementary Material]

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

(a) Model (2.3), $d = 200$     (b) Model (2.4), $d = 200$     (c) Model (2.5), $d = 200$

(d) Model (2.3), $d = 400$     (e) Model (2.4), $d = 400$     (f) Model (2.5), $d = 400$

Figure 2: Simulation results for the three examples considered in §2, in two different settings for the dimension $d = 200, 400$. Here the parameter $\theta \approx \frac{n}{s^2 \log d}$ describes the relationship between sample size, dimension and sparsity of the signal. Algorithm 2 dominates in most settings, with exceptions when $\theta$ is too small, in which case none of the approaches provides meaningful results.

worse than the estimate produced by Algorithm 2, and it needs an initialization (for the first step of Algorithm 1 is used) and further requires a rough knowledge of the sparsity $s$, whereas both Algorithms 1 and 2 do not require an estimate of $s$.

## 5 Discussion

In this paper we proposed a two-step procedure for estimation of MPR models with standard Gaussian designs. We argued that the MPR models form a rich class including numerous additive SIMs (i.e., $Y = h(\boldsymbol{X}^\top \boldsymbol{\beta}^*) + \varepsilon$) with an even and increasing on $\mathbb{R}^+$ link function $h$. Our algorithm is based solely on convex optimization, and achieves optimal rates of estimation.

Our procedure does require that the sample size $n \gtrsim s^2 \log d$ to ensure successful initialization. The same condition has been exhibited previously, e.g., in [7] for the phase retrieval model, and in works on sparse principal components analysis [see, e.g., 3, 15, 33]. We anticipate that for a certain subclass of MPR models, the sample size requirement $n \gtrsim s^2 \log d$ is necessary for computationally efficient algorithms to exist. We conjecture that models (2.3)-(2.5) are such models. It is however certainly not true that this sample size requirement holds for all models from the MPR class. For example, the following model can be solved efficiently by applying the Lasso algorithm, without requiring the sample size scaling $n \gtrsim s^2 \log d$

$$Y = \text{sign}(\boldsymbol{X}^\top \boldsymbol{\beta}^* + c),$$

where $c < 0$ is fixed. This discussion leads to the important question under what conditions of the (known) link and error distribution $(f, \varepsilon)$ one can efficiently solve the SIM $Y = f(\boldsymbol{X}^\top \boldsymbol{\beta}^*, \varepsilon)$ with an optimal sample complexity. We would like to investigate this issue further in our future work.

**Acknowledgments:** The authors would like to thank the reviewers and meta-reviewers for carefully reading the manuscript and their helpful suggestions which improved the presentation. The authors would also like to thank Professor Xiaodong Li for kindly providing the code for the TWF algorithm.

## Footnotes

[1] In such applications it is typically assumed that $\boldsymbol{X} \in \mathbb{C}^d$ is a complex normal random vector. In this paper for simplicity we only consider the real case $\boldsymbol{X} \in \mathbb{R}^d$.

[2]The fact that $\mathbb{E}\overline{\partial}_\varepsilon h(z+\varepsilon)$ exists is implied by $\mathbb{E}|h(z+\varepsilon)| < \infty$.

[3]Note also that $\varepsilon \mapsto \overline{\partial}_\varepsilon h(z+\varepsilon)$ is monotone and hence measurable.

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

# A  Notation

In addition to the notation defined in §1, throughout the appendices we use $\odot$ to denote the Hadamard (or element-wise) product, and dot product will sometimes be denoted with angle notation $\langle \cdot, \cdot \rangle$, to facilitate the display of long equations. For a matrix $\mathbf{A}$ we denote the max and $\ell_2$ norms with $\|\mathbf{A}\|_{\max} = \max_{i,j} |A_{ij}|$ and $\|\mathbf{A}\|_2 = \sup_{\|\mathbf{v}\|_2 = 1} \|\mathbf{A}\mathbf{v}\|_2$ respectively. If $\mathbf{A}$ is symmetric we denote its spectrum ordered in decreasing manner by $\lambda_j(\mathbf{A})$.

# B  Auxiliary Results

Here we collect several results which we use in the later development.

**Lemma B.1** (Lemma 5 [2])**.**  Consider the following optimization program:

$$\widehat{\mathbf{Z}} = \operatorname*{argmax}_{\operatorname{tr}(\mathbf{Z})=1, \mathbf{Z} \in \mathbb{S}^d_+} \operatorname{tr}(\mathbf{A}\mathbf{Z}) - \lambda_n \sum_{i,j=1}^d |Z_{ij}|, \tag{B.1}$$

where $\mathbb{S}^d_+$ is the set of all the $d \times d$ positive semi-definite matrices. Suppose there exists a matrix $\mathbf{U}$ (independent of $\widehat{\mathbf{z}}$) satisfying:

$$U_{ij} = \begin{cases} \operatorname{sign}(\widehat{z}_i) \operatorname{sign}(\widehat{z}_j), & \text{if } \widehat{z}_i \widehat{z}_j \neq 0; \\ \in [-1, 1], & \text{otherwise.} \end{cases} \tag{B.2}$$

Then if $\widehat{\mathbf{z}}$ is the principal eigenvector of the matrix $\mathbf{A} - \lambda_n \mathbf{U}$, $\widehat{\mathbf{z}}\widehat{\mathbf{z}}^\top$ is the optimal solution to problem (B.1).

For convenience of the reader we briefly recall the notation and result on Gaussian concentration of non-Lipschitz functions used by [1], which we apply in Lemma E.8 below. For the set $[\ell]$, we denote with $P_\ell$ the set of its partitions into non-empty and non-intersecting disjoint sets. For a partition $\mathcal{J} = \{J_1, \ldots, J_k\}$, and an $\ell$-indexed matrix $\mathbf{A} = (a_{\mathbf{i}})_{\mathbf{i} \in [n]^\ell}$, define the norm:

$$\|\mathbf{A}\|_{\mathcal{J}} = \sup \Big\{ \sum_{\mathbf{i} \in [n]^\ell} a_{\mathbf{i}} \prod_{l=1}^k x^{(l)}_{\mathbf{i}_{J_l}} : \|x^{(l)}_{\mathbf{i}_{J_l}}\|_2 \leq 1, 1 \leq l \leq k \Big\},$$

where the indexing should be understood as $\mathbf{i}_I := (i_k)_{k \in I}$. Given the convention that $\#\mathcal{J} = |\mathcal{J}|$ is the cardinality of the set $\mathcal{J}$ we restate (a shortened) version of Theorem 1.4 of [1].

**Theorem B.2** (Theorem 1.4 [1])**.**  Let $\boldsymbol{X} = (X_1, \ldots, X_n)$ be a random vector with independent components, such that for all $i \leq n$, $\|X_i\|_{\psi_2} \leq \Upsilon$. Then for every polynomial $f : \mathbb{R}^n \mapsto \mathbb{R}$ of degree $L$ and every $p \geq 2$ we have:

$$\|f(X) - \mathbb{E}f(X)\|_p \leq K_L \sum_{\ell=1}^L \Upsilon^\ell \sum_{\mathcal{J} \in P_\ell} p^{\#\mathcal{J}/2} \|\mathbb{E}\mathbf{D}^\ell f(X)\|_{\mathcal{J}}.$$

Here $\mathbf{D}^\ell$ is the $\ell^{\text{th}}$ derivative of $f$.

# C  Preliminary Proofs

*Proof of Proposition 2.2.*  We have the following equality:

$$c_0 = \operatorname{Cov}(f(Z, \varepsilon), Z^2) = \mathbb{E}[\varphi(Z)Z^2] - \mathbb{E}\varphi(Z) = \mathbb{E}D^2\varphi(Z) > 0,$$

where the last (and key) equation follows by Stein's Lemma [see, e.g., Lemma 4 of 30]. $\square$

*Proof of Lemma 3.1.*  First of all we notice that:

$$\mathbb{E}[Y(\boldsymbol{X}^{\otimes 2} - \mathbf{I})] = \mathbb{E}[(Y - \mu)(\boldsymbol{X}^{\otimes 2} - \mathbf{I})] = \mathbb{E}(Y - \mu)\boldsymbol{X}^{\otimes 2}.$$

Hence proving Lemma 3.1 is equivalent to showing:

$$\boldsymbol{\beta}^* = \operatorname*{argmax}_{\|\mathbf{v}\|_2 = 1} \mathbb{E}[(Y - \mu)(\mathbf{v}^\top \boldsymbol{X})^2].$$

Next, decompose:

$$\mathbf{v}^\top \boldsymbol{X} = (\mathbf{v}^\top \boldsymbol{\beta}^*)\boldsymbol{\beta}^{*\top}\boldsymbol{X} + (\mathbf{v} - (\mathbf{v}^\top \boldsymbol{\beta}^*)\boldsymbol{\beta}^{*\top})\boldsymbol{X} = (\mathbf{v}^\top \boldsymbol{\beta}^*)\boldsymbol{\beta}^{*\top}\boldsymbol{X} + \boldsymbol{\beta}^{\perp\top}\boldsymbol{X},$$

where we used the shorthand notation $\boldsymbol{\beta}^\perp := \mathbf{v} - (\mathbf{v}^\top\boldsymbol{\beta}^*)\boldsymbol{\beta}^*$, for the vector $\boldsymbol{\beta}^\perp$ which is orthogonal to $\boldsymbol{\beta}$. In terms of this notation, we have the following identity:

$$\mathbb{E}[(Y-\mu)(\mathbf{v}^\top\boldsymbol{X})^2] = (\mathbf{v}^\top\boldsymbol{\beta}^*)^2\mathbb{E}[(Y-\mu)(\boldsymbol{\beta}^{*\top}\boldsymbol{X})^2]$$
$$+ 2(\mathbf{v}^\top\boldsymbol{\beta}^*)\mathbb{E}[(Y-\mu)(\boldsymbol{\beta}^{*\top}\boldsymbol{X})(\boldsymbol{\beta}^{\perp\top}\boldsymbol{X})] + \mathbb{E}[(Y-\mu)(\boldsymbol{\beta}^{\perp\top}\boldsymbol{X})^2].$$

We next deal with the last two terms of the above decomposition. Since $\boldsymbol{\beta}^{\perp\top}\boldsymbol{X} \perp\!\!\!\perp \boldsymbol{\beta}^{*\top}\boldsymbol{X}$ we have that the second term:

$$\mathbb{E}[(Y-\mu)(\boldsymbol{\beta}^{*\top}\boldsymbol{X})(\boldsymbol{\beta}^{\perp\top}\boldsymbol{X})] = \mathbb{E}[(Y-\mu)(\boldsymbol{\beta}^{*\top}\boldsymbol{X})]\mathbb{E}[\boldsymbol{\beta}^{\perp\top}\boldsymbol{X}] = 0.$$

For the third term due to the same independence ($\boldsymbol{\beta}^{\perp\top}\boldsymbol{X} \perp\!\!\!\perp \boldsymbol{\beta}^{*\top}\boldsymbol{X}$) we have:

$$\mathbb{E}[(Y-\mu)(\boldsymbol{\beta}^{\perp\top}\boldsymbol{X})^2] = \mathbb{E}[(Y-\mu)]\mathbb{E}[(\boldsymbol{\beta}^{\perp\top}\boldsymbol{X})^2] = 0.$$

Hence:

$$\mathbb{E}[(Y-\mu)(\mathbf{v}^\top\boldsymbol{X})^2] = (\mathbf{v}^\top\boldsymbol{\beta}^*)^2\mathbb{E}[(Y-\mu)(\boldsymbol{\beta}^{*\top}\boldsymbol{X})^2] = (\mathbf{v}^\top\boldsymbol{\beta}^*)^2 c_0.$$

Since (2.1) implies that $c_0 > 0$, by Cauchy-Schwartz the maximizer of the above expression is $\mathbf{v} = \pm\boldsymbol{\beta}^*$. $\qquad\square$

*Proof of Proposition 2.3.* We prove the three statements in turn.

(i) Let $Z'$ be an independent copy of $Z$. We have the following chain of equalities:
$$c_0 = \mathbb{E}(\varphi(Z) - \mathbb{E}\varphi(Z))Z^2 = \mathbb{E}(\varphi(Z) - \mathbb{E}\varphi(Z'))Z^2 = \mathbb{E}(\varphi(Z) - \varphi(Z'))Z^2.$$

By symmetry one also has: $c_0 = \mathbb{E}(\varphi(Z') - \varphi(Z))(Z')^2$. Adding the last two equations yields:
$$2c_0 = \mathbb{E}[(\varphi(Z) - \varphi(Z'))(Z^2 - (Z')^2)]$$
$$= \mathbb{E}_{X,X'\sim|\mathcal{N}(0,1)|}[(\varphi(X) + \varphi(-X) - (\varphi(X') + \varphi(-X')))(X^2 - (X')^2)]/2$$
$$> 0,$$

where we used the fact that $\mathrm{sign}(\varphi(X) + \varphi(-X) - (\varphi(X') + \varphi(-X'))) = \mathrm{sign}(X^2 - (X')^2)$. The last inequality is strict since by our condition the integrand is strictly positive on the set $[0, z_2] \times [z_1, \infty) \subset \mathbb{R}^2$ which is a set of positive Lebesgue measure.

(ii) To see (ii) for any two points $x < y$, take $v_x \in \partial\varphi(x)$ and $v_y \in \partial\varphi(y)$ to be arbitrary points in the corresponding sub-differentials. Adding the following two inequalities:
$$\varphi(x) - \varphi(y) \geq v_y(x - y), \quad \varphi(y) - \varphi(x) \geq v_x(y - x),$$
we conclude that $(v_x - v_y)(x - y) \geq 0$. Notice that since $\varphi$ is convex, by Jensen's inequality, $\varphi(z) + \varphi(-z) \geq 2\varphi(0)$ for any $z \geq 0$. Next take $z > z' > 0$, and consider the difference:
$$\varphi(z) + \varphi(-z) - \varphi(z') - \varphi(-z') \geq (v_{z'} - v_{-z'})(z - z') \geq 0,$$
where $v_{z'} \in \partial\varphi(z')$ and $v_{-z'} \in \partial\varphi(-z')$ are arbitrary sub-gradients, and the last inequality follows by the fact that $z' > 0$ and hence $v_{z'} \geq v_{-z'}$ as we verified before. The above inequality becomes strict whenever $z' \geq z_0$ since $v_{z'} - v_{-z'}$ is non-decreasing and by assumption:
$$z_0 v_{z_0} \geq \varphi(z_0) - \varphi(0) > \varphi(0) - \varphi(-z_0) \geq v_{-z_0} z_0,$$
and hence $v_{z_0} - v_{-z_0} > 0$. Hence we may take $z_1 = 2z_0$ and $z_2 = z_0$ in (i) to complete the proof.

(iii) Statement (iii) is an implication of the fact that we can control the tail bound of $g_1(Z)$. Notice that when $0 < t \leq \max\{|a|, |b|\}$ we trivially have $\mathbb{P}(|g_1(Z)| \geq t) \leq 1$. When $t > \max\{|a|, |b|\}$ using our assumption, by a standard normal tail bound we have:
$$\mathbb{P}(|g_1(Z)| \geq t) \leq \mathbb{P}(|Z| \geq \sqrt{t/C}) \leq 2\exp(-t/(2C)) \leq \exp(1 - t/(2C)).$$
Hence setting $K = \max\{|a|, |b|, 2C\}$ shows that in any case $\mathbb{P}(|g_1(Z)| \geq t) \leq \exp(1 - t/K)$, which shows that $\|g_1(Z)\|_{\psi_1} \leq cK < \infty$ for some absolute constant $c$. Finally by the triangle inequality we conclude:
$$\|g_1(Z)\|_{\psi_1} + \|g_2(\varepsilon)\|_{\psi_1} < \infty,$$
which completes the proof.

$\qquad\square$

**Lemma C.1.** *If $h$ is a convex function such that $h(z_0) + h(-z_0) > 2h(0)$ for some $z_0 > 0$, $\mathbb{E}|h(z + \varepsilon)| < \infty$ for every $z \in \mathbb{R}$, and $\mathbb{E}|h(Z + \varepsilon)| < \infty$ we have $\varphi(z) = \mathbb{E}h(z + \varepsilon)$ is convex, sub-differentiable and there exists a $z_0' > 0$ such that $\varphi(z_0') + \varphi(-z_0') > 2\varphi(0)$.*

*Proof of Lemma C.1.* Since the function $h$ is convex and the expectation is a linear operator it follows that $\varphi(z)$ is indeed convex. The linearity of the expectation operator, coupled with the fact that the function $|h(z+\varepsilon)|$ is integrable for all $z$, additionally implies that $\varphi(z)$ is sub-differentiable with $\mathbb{E}\overline{\partial}_\varepsilon h(z+\varepsilon) \in \partial\varphi(z)^2$, where $\overline{\partial}_\varepsilon h(z+\varepsilon) \in \partial h(z+\varepsilon)$ is chosen so that $\varepsilon \mapsto \overline{\partial}_\varepsilon h(z+\varepsilon)$ is a function[3]. Next, notice that for any fixed $\varepsilon$, we have:

$$\mathbb{E}_Z[h(Z+\varepsilon) + h(-Z+\varepsilon)] > 2h(\varepsilon).$$

The last inequality is strict, since by Jensen's inequality $\mathbb{E}_Z[h(Z+\varepsilon) + h(-Z+\varepsilon)] \geq h(\mathbb{E}Z + \varepsilon) + h(-\mathbb{E}Z + \varepsilon) = 2h(\varepsilon)$, and equality can be achieved only when $h$ is linear, which is not the case since $h(z_0) + h(-z_0) > 2h(0)$ by assumption. Take an expectation with respect to $\varepsilon$ and exchange the expectations (by Fubini's theorem, recall that $\mathbb{E}|h(Z+\varepsilon)| < \infty$) to obtain:

$$\mathbb{E}_Z\mathbb{E}_\varepsilon[h(Z+\varepsilon) + h(-Z+\varepsilon)] > 2\mathbb{E}_\varepsilon h(\varepsilon).$$

Naturally, the above implies the existence of $z_0'$ such that:

$$\mathbb{E}_\varepsilon[h(z_0'+\varepsilon) + h(-z_0'+\varepsilon)] > 2\mathbb{E}_\varepsilon h(\varepsilon),$$

and completes the proof. $\qquad\square$

# D   Proofs for Initialization Step

*Proof of Proposition 3.2.* The proof follows by an application of Lemma B.1 and Lemma D.2. $\quad\square$

**Lemma D.1.** Let $\mathbf{A} = a\mathbf{v}\mathbf{v}^\top - b\mathbf{w}\mathbf{w}^\top$ be a symmetric rank two matrix, with $a > b \geq 0$ and $\|\mathbf{v}\|_2 = \|\mathbf{w}\|_2 = 1$, and let $\mathbf{N}$ be a symmetric noise matrix. Then, assuming that $\|\mathbf{N}\|_2 \leq \frac{a-b}{2}$ the principal eigenvector $\widehat{\mathbf{v}}$ of $\mathbf{A} + \mathbf{N}$ satisfies:

$$|\widehat{\mathbf{v}}^\top \mathbf{v}| \geq \left[\frac{a - b - 2\|\mathbf{N}\|_2}{a}\right]^{1/2}$$

*Proof of Lemma D.1.* First off, an elementary calculation shows that the non-zero spectrum of $\mathbf{A}$ is:

$$\{\lambda_1(\mathbf{A}), \lambda_d(\mathbf{A})\} = \left\{\frac{a - b \pm \sqrt{(a-b)^2 + 4ab(1 - \mathbf{v}^\top\mathbf{w})}}{2}\right\}.$$

Next we have:

$$a(\mathbf{v}^\top\widehat{\mathbf{v}})^2 + \|\mathbf{N}\|_2 \geq \widehat{\mathbf{v}}^\top(\mathbf{A}+\mathbf{N})\widehat{\mathbf{v}} \geq \lambda_1(\mathbf{A}) - \|\mathbf{N}\|_2,$$

and hence:

$$(\mathbf{v}^\top\widehat{\mathbf{v}})^2 \geq \frac{a - b + \sqrt{(a-b)^2 + 4ab(1 - \mathbf{v}^\top\mathbf{w})}}{2a} - 2\frac{\|\mathbf{N}\|_2}{a} \tag{D.1}$$

$$\geq \frac{a - b - 2\|\mathbf{N}\|_2}{a}, \tag{D.2}$$

where the last inequality follows by Cauchy-Schwartz. $\qquad\square$

**Lemma D.2.** Assume that $n$ is large enough so that $s\sqrt{\frac{\log d}{n}} < (\frac{1}{6} - \frac{\kappa}{4})\frac{c_0}{(C_1+C_2)}$ for some small but fixed $\kappa > 0$ and constants $C_1, C_2$ as defined in Lemmas D.6 and D.7. Put $\lambda_n = (C_1 + C_2)\sqrt{\frac{\log d}{n}}$. There exists a sign matrix $\widehat{\mathbf{U}}$ with $\widehat{\mathbf{U}}_{S_{\boldsymbol{\beta}^*} S_{\boldsymbol{\beta}^*}} = \text{sign}(\boldsymbol{\beta}^*_{S_{\boldsymbol{\beta}^*}})\,\text{sign}(\boldsymbol{\beta}^*_{S_{\boldsymbol{\beta}^*}})^\top$ such that the principal eigenvector of $\widehat{\boldsymbol{\Sigma}} - \lambda\widehat{\mathbf{U}}$, $\widehat{\mathbf{v}}$ satisfies:

$$|\widehat{\mathbf{v}}^\top\boldsymbol{\beta}^*| \geq \kappa,$$

with probability at least $1 - 4d^{-1} - O(n^{-1})$.

**Remark D.3.** The proof of Lemma D.2 also shows that with high probability the vector $\widehat{\mathbf{v}}$ can be identified with a vector $\widetilde{\mathbf{v}}$ (the principal eigenvector of $\widehat{\boldsymbol{\Sigma}}_{S_{\boldsymbol{\beta}^*}, S_{\boldsymbol{\beta}^*}} - \lambda_n\widehat{\mathbf{U}}_{S_{\boldsymbol{\beta}^*}, S_{\boldsymbol{\beta}^*}}$ see below) which is independent of the data $\mathbf{X}_{S^c_{\boldsymbol{\beta}^*}}$ such that $\widehat{\mathbf{v}} \equiv \widetilde{\mathbf{v}}$ with high probability. This becomes evident upon realizing that the matrix $\mathbf{N}_{S_{\boldsymbol{\beta}^*} S_{\boldsymbol{\beta}^*}}$ depends solely on $\mathbf{X}_{S_{\boldsymbol{\beta}^*}}$. In addition it is evident that the support $\text{supp}(\widehat{\mathbf{v}}) \subset S_{\boldsymbol{\beta}^*}$ and $\text{supp}(\widetilde{\mathbf{v}}) \subset S_{\boldsymbol{\beta}^*}$.

*Proof of Lemma D.2.* Setting $\lambda_n = (C_1 + C_2)\sqrt{\frac{\log d}{n}}$ and using Lemma D.4 gives us that

$$\widehat{\boldsymbol{\Sigma}} - \lambda_n \widehat{\mathbf{U}} = \left[ \begin{array}{c|c} \eta \boldsymbol{\beta}^*_{S_{\boldsymbol{\beta}^*}} \boldsymbol{\beta}^{*\top}_{S_{\boldsymbol{\beta}^*}} - \lambda_n \operatorname{sign}(\boldsymbol{\beta}^*_{S_{\boldsymbol{\beta}^*}}) \operatorname{sign}(\boldsymbol{\beta}^*_{S_{\boldsymbol{\beta}^*}})^\top + \mathbf{N}_{S_{\boldsymbol{\beta}^*} S_{\boldsymbol{\beta}^*}} & \mathbf{N}_{S_{\boldsymbol{\beta}^*} S^c_{\boldsymbol{\beta}^*}} - \lambda_n \widehat{\mathbf{U}}_{S_{\boldsymbol{\beta}^*} S^c_{\boldsymbol{\beta}^*}} \\ \hline \mathbf{N}_{S^c_{\boldsymbol{\beta}^*} S_{\boldsymbol{\beta}^*}} - \lambda_n \widehat{\mathbf{U}}_{S^c_{\boldsymbol{\beta}^*} S_{\boldsymbol{\beta}^*}} & \mathbf{N}_{S^c_{\boldsymbol{\beta}^*} S^c_{\boldsymbol{\beta}^*}} - \lambda_n \widehat{\mathbf{U}}_{S^c_{\boldsymbol{\beta}^*} S^c_{\boldsymbol{\beta}^*}} \end{array} \right].$$

We can select the sign matrix $\widehat{\mathbf{U}}$ such that all three terms which do not correspond to the $S_{\boldsymbol{\beta}^*} S_{\boldsymbol{\beta}^*}$ "corner" of the above visualization are $\equiv 0$, since by Lemma D.4 we have that $\|\mathbf{N}\|_{\max} \leq (C_1 + C_2)\sqrt{\frac{\log d}{n}} \leq \lambda_n$ with high probability. Recall that by our assumption on the sample size we have $\lambda_n \leq \frac{c_0}{6s}$ and hence $\lambda_n \widehat{\mathbf{U}}_{S^c_{\boldsymbol{\beta}^*} S_{\boldsymbol{\beta}^*}} \leq \frac{c_0}{6} \frac{\operatorname{sign}(\boldsymbol{\beta}^*_{S_{\boldsymbol{\beta}^*}})}{\sqrt{s}} \frac{\operatorname{sign}(\boldsymbol{\beta}^*_{S_{\boldsymbol{\beta}^*}})^\top}{\sqrt{s}}$. Using Lemmas D.1 and D.4 on the event $\|\mathbf{N}_{S_{\boldsymbol{\beta}^*} S_{\boldsymbol{\beta}^*}}\|_2 \leq (C_1 + C_2)s\sqrt{\frac{\log d}{n}}$ we have:

$$|\widehat{\mathbf{v}}^\top_{S_{\boldsymbol{\beta}^*}} \boldsymbol{\beta}^*_{S_{\boldsymbol{\beta}^*}}| \geq \frac{\eta - \frac{c_0}{6} - 2\|\mathbf{N}_{S_{\boldsymbol{\beta}^*} S_{\boldsymbol{\beta}^*}}\|_2}{\eta} \geq$$

$$\geq 1 - \frac{1}{3} - 4\|\mathbf{N}_{S_{\boldsymbol{\beta}^*} S_{\boldsymbol{\beta}^*}}\|_2 / c_0 \geq \frac{2}{3} - 4\frac{(C_1 + C_2)}{c_0} s\sqrt{\frac{\log d}{n}} \geq \kappa,$$

for values of $n$ large enough so that the above expression is positive, which concludes the proof. $\square$

**Lemma D.4.** We have that:

$$\widehat{\boldsymbol{\Sigma}} = \eta \boldsymbol{\beta}^* \boldsymbol{\beta}^{*\top} + \mathbf{N}, \tag{D.3}$$

where $\eta > c_0/2$ and $\|\mathbf{N}_{S_{\boldsymbol{\beta}^*} S_{\boldsymbol{\beta}^*}}\|_2 \leq (C_1 + C_2)s\sqrt{\frac{\log d}{n}}$ and $\|\mathbf{N}\|_{\max} \leq (C_1 + C_2)\sqrt{\frac{\log d}{n}}$ with probability at least $1 - 4d^{-1} - O(n^{-1})$, where $C_1$ and $C_2$ are constants depending on $f, \varepsilon$.

*Proof of Lemma D.4.* First we observe that decomposition (D.3) holds with:

$$\eta \boldsymbol{\beta}^* \boldsymbol{\beta}^{*\top} = \frac{1}{n}\sum_{i=1}^n Y_i (\boldsymbol{\beta}^{*\top} \boldsymbol{X}_i)^2 \boldsymbol{\beta}^* \boldsymbol{\beta}^{*\top} + \frac{1}{n}\sum_{i=1}^n Y_i (\mathbf{P}_{\boldsymbol{\beta}^{*\perp}} - \mathbf{I}_d),$$

$$\mathbf{N} = \frac{1}{n}\sum_{i=1}^n Y_i (\boldsymbol{\beta}^{*\top} \boldsymbol{X}_i)(\boldsymbol{\beta}^* \boldsymbol{X}_i^\top \mathbf{P}_{\boldsymbol{\beta}^{*\perp}} + \mathbf{P}_{\boldsymbol{\beta}^{*\perp}} \boldsymbol{X}_i \boldsymbol{\beta}^{*\top}) + \frac{1}{n}\sum_{i=1}^n Y_i [\mathbf{P}_{\boldsymbol{\beta}^{*\perp}} (\boldsymbol{X}_i^{\otimes 2} - \mathbf{I}_d) \mathbf{P}_{\boldsymbol{\beta}^{*\perp}}],$$

where $\mathbf{P}_{\boldsymbol{\beta}^{*\perp}} = (\mathbf{I}_d - \boldsymbol{\beta}^* \boldsymbol{\beta}^{*\top})$. Lemma D.5 shows that $\eta \geq c_0/2$ with probability at least $1 - O(n^{-1})$. Next, Lemma D.6 and Lemma D.7 show that:

$$\|\mathbf{N}\|_{\max} \leq (C_1 + C_2)\sqrt{\frac{\log d}{n}},$$

with probability at least $1 - 4d^{-1} - O(n^{-1})$, where the constants $C_1$ and $C_2$ depend on $f, \varepsilon$. Using the fact that $\|\mathbf{N}_{S_{\boldsymbol{\beta}^*} S_{\boldsymbol{\beta}^*}}\|_2 \leq \|\mathbf{N}_{S_{\boldsymbol{\beta}^*} S_{\boldsymbol{\beta}^*}}\|_1 \leq s\|\mathbf{N}\|_{\max}$ completes the proof. $\square$

**Lemma D.5.** We have that $\eta$ defined in (D.3) satisfies

$$\eta \geq c_0/2,$$

with probability at least $1 - \frac{4 \operatorname{Var}[f(Z, \varepsilon)(Z^2 - 1)]}{c_0^2} n^{-1}$.

*Proof of Lemma D.5.* Grouping the first two terms by Chebyshev's inequality we have that:

$$\mathbb{P}\left( \left| \frac{1}{n}\sum_{i=1}^n Y_i ((\boldsymbol{\beta}^{*\top} \boldsymbol{X}_i)^2 - 1) - c_0 \right| \geq t \right) \leq \frac{\operatorname{Var}[f(Z, \varepsilon)(Z^2 - 1)]}{nt^2}.$$

Notice that in the last inequality we have $\operatorname{Var}[(f(Z, \varepsilon)(Z^2 - 1)] < \infty$ since we are assuming that $f(Z, \varepsilon)$ is sub-exponential. Setting $t = c_0/2$ brings the above probability bound to zero at a rate $n^{-1}$.

$\square$

**Lemma D.6.** We have that:

$$\left\| \frac{1}{n}\sum_{i=1}^n Y_i (\boldsymbol{\beta}^{*\top} \boldsymbol{X}_i)[\boldsymbol{\beta}^* \boldsymbol{X}_i^\top \mathbf{P}_{\boldsymbol{\beta}^{*\perp}} + \mathbf{P}_{\boldsymbol{\beta}^{*\perp}} \boldsymbol{X}_i \boldsymbol{\beta}^{*\top}] \right\|_{\max} \leq C_1 \sqrt{\frac{\log d}{n}},$$

where $C_1$ is a constant depending on $f, \varepsilon$, with probability at least $1 - 2d^{-1} - \frac{\mathrm{Var}[f^2(Z,\varepsilon)Z^2]}{(\mathbb{E}[f^2(Z,\varepsilon)Z^2])^2}n^{-1}$.

*Proof of Lemma D.6.* We will only deal with the first term of the sum, as the second term follows by the same argument after transposition. First notice that $Y_i(\boldsymbol{\beta}^{*\top}\boldsymbol{X}_i) \perp\!\!\!\perp \boldsymbol{X}_i^\top \mathbf{P}_{\boldsymbol{\beta}^{*\perp}}$. Analyzing the first part of this term row-wise, for $j \in S_{\boldsymbol{\beta}^*}$ ($\beta_j^* \neq 0$) we have:

$$\frac{1}{n}\sum_{i=1}^n Y_i\beta_j^*(\boldsymbol{\beta}^{*\top}\boldsymbol{X}_i)\boldsymbol{X}_i^\top\mathbf{P}_{\boldsymbol{\beta}^{*\perp}} \sim \mathcal{N}\Big(0, \beta_j^{*2}\frac{1}{n^2}\sum_{i=1}^n Y_i^2(\boldsymbol{\beta}^{*\top}\boldsymbol{X}_i)^2\mathbf{P}_{\boldsymbol{\beta}^{*\perp}}\Big).$$

By Chebyshev's inequality we have:

$$\mathbb{P}\Big(\Big|\frac{1}{n}\sum_{i=1}^n Y_i^2(\boldsymbol{\beta}^{*\top}\boldsymbol{X}_i)^2 - \mathbb{E}[f^2(Z,\varepsilon)Z^2]\Big| \geq t\Big) \leq \frac{\mathrm{Var}[f^2(Z,\varepsilon)Z^2]}{nt^2},$$

assuming $\mathrm{Var}[f^2(Z,\varepsilon)Z^2] < \infty$. Putting $t = \mathbb{E}[f^2(Z,\varepsilon)Z^2]$ brings the above probability converge to zero at a rate $n^{-1}$. Hence, by a conditioning argument, a standard normal tail bound coupled with the facts that $\|\mathbf{P}_{\boldsymbol{\beta}^{*\perp}}\|_2 \leq 1, |\beta_j^*| \leq 1$ and a union bound, we obtain:

$$\mathbb{P}\Big(\max_{j\in[d]}\Big\|\frac{1}{n}\sum_{i=1}^n Y_i(\boldsymbol{\beta}^{*\top}\boldsymbol{X}_i)\beta_j^*\boldsymbol{X}_i^\top\mathbf{P}_{\boldsymbol{\beta}^{*\perp}}\Big\|_\infty > t\Big) \leq 2d^2\exp\Big(-\frac{nt^2}{4\mathbb{E}[f^2(Z,\varepsilon)Z^2]}\Big),$$

Plugging in

$$t = 2\sqrt{3\mathbb{E}f^2(Z,\varepsilon)Z^2}\sqrt{\frac{\log d}{n}},$$

brings the probability to $2d^{-1}$. This completes the proof with $C_1 = 4\sqrt{3\mathbb{E}f^2(Z,\varepsilon)Z^2}$. $\qquad\square$

**Lemma D.7.** Let $Y_i = f(\boldsymbol{X}_i^\top\boldsymbol{\beta}^*, \varepsilon)$, where $\boldsymbol{X}_i \sim \mathcal{N}(0, \mathbf{I})$. Assume that $f$ and $\varepsilon$ are such that $\|f(Z,\varepsilon)\|_{\psi_1} \leq K$ for $Z \sim \mathcal{N}(0,1)$ and $Z \perp\!\!\!\perp \varepsilon$, and in addition let $\log d = o(n/\log^2 n)$. Then:

$$\Big\|\frac{1}{n}\sum_{i=1}^n Y_i\mathbf{P}_{\boldsymbol{\beta}^{*\perp}}(\boldsymbol{X}_i^{\otimes 2} - \mathbf{I}_d)\mathbf{P}_{\boldsymbol{\beta}^{*\perp}}\Big\|_{\max} \leq \sqrt{\frac{C_2\log d}{n}},$$

with probability at least $1 - 2d^{-1} - (2^{11}K^4/(\mathbb{E}f^2(Z,\varepsilon))^2 + e)n^{-1}$, for some absolute value $C_2$ depending on $K$, and large values of $n$.

*Proof of Lemma D.7.* Notice that by the properties of the multivariate normal distribution one has that $Y_i \perp\!\!\!\perp \mathbf{P}_{\boldsymbol{\beta}^{*\perp}}(\boldsymbol{X}_i^{\otimes 2}-\mathbf{I}_d)\mathbf{P}_{\boldsymbol{\beta}^{*\perp}}$. Next we have that $\boldsymbol{Z}_i := \mathbf{P}_{\boldsymbol{\beta}^{*\perp}}\boldsymbol{X}_i \sim \mathcal{N}(0, \mathbf{P}_{\boldsymbol{\beta}^{*\perp}})$, and thus, since $\|\mathbf{P}_{\boldsymbol{\beta}^{*\perp}}\|_2 \leq 1$, we have that each individual entry of $\boldsymbol{Z}_i$ is a normally distributed random variable with variance at most one. Hence we have that for any $j, k \in [d]$: $\|\boldsymbol{Z}_{ij}\boldsymbol{Z}_{ik}\|_{\psi_1} \leq 2\|\boldsymbol{Z}_{ij}\|_{\psi_2}\|\boldsymbol{Z}_{ik}\|_{\psi_2} \leq 2$, and hence conditionally on $Y_i$ one has

$$\|\boldsymbol{Z}_{ij}\boldsymbol{Z}_{ik} - \mathbb{E}\boldsymbol{Z}_{ij}\boldsymbol{Z}_{ik}\|_{\psi_1} = \|\boldsymbol{Z}_{ij}\boldsymbol{Z}_{ik} - \mathbf{P}_{\boldsymbol{\beta}^{*\perp},jk}\|_{\psi_1} \leq 4,$$

for all $j, k \in [d]$. Next conditionally on the $Y_i$ values and a Bernstein type of inequality (see, e.g., Proposition 5.16 of [32]) we obtain:

$$\mathbb{P}\Big(\Big\|\frac{1}{n}\sum_{i=1}^n Y_i\mathbf{P}_{\boldsymbol{\beta}^{*\perp}}(\boldsymbol{X}_i^{\otimes 2} - \mathbf{I}_d)\mathbf{P}_{\boldsymbol{\beta}^{*\perp}}\Big\|_{\max} \geq t\Big) \tag{D.4}$$

$$\leq 2d^2\exp\Big[-c\min\Big(\frac{nt^2}{16n^{-1}\sum_{i=1}^n Y_i^2}, \frac{nt}{4\max_{i\in[n]}|Y_i|}\Big)\Big], \tag{D.5}$$

for an absolute constant $c > 0$. Using the union bound and the fact that $Y_i$ are sub-exponential we obtain:

$$\mathbb{P}(\max|Y_i| \geq t) \leq n\exp(1 - t/(c'K)), \tag{D.6}$$

for some absolute constant. Setting $t = 2c'K\log(n)$ brings the above probability converging to zero at a rate $n^{-1}$. Furthermore by Chebyshev's inequality we obtain:

$$\mathbb{P}\Big(\Big|n^{-1}\sum_{i=1}^n Y_i^2 - \mathbb{E}Y^2\Big| \geq t\Big) \leq \mathrm{Var}(Y_i^2)n^{-1}t^{-2} \leq 2^9K^4n^{-1}t^{-2}, \tag{D.7}$$

and thus we can set $t = \mathbb{E}Y^2/2$ to bring the above probability to zero at a rate $n^{-1}$. In addition we have $\mathbb{E}Y^2/2 \leq n^{-1}\sum_{i=1}^n Y_i^2 \leq 2\mathbb{E}Y^2$ with probability at least $2^{11}K^4n^{-1}/(\mathbb{E}Y^2)^2$. Selecting

$t = \sqrt{\frac{96\mathbb{E}Y^2 \log d}{cn}}$ in (D.4) gives us that:

$$t \leq \frac{\mathbb{E}Y^2}{c'K \log n} \leq \frac{16n^{-1}\sum_{i=1}^{n} Y_i^2}{4 \max_{i \in [n]} |Y_i|},$$

where the first inequality in the preceding display holds for large enough values of $n$ so long as $\log d = o(n/\log^2 n)$. Hence we conclude:

$$\left\| \frac{1}{n} \sum_{i=1}^{n} Y_i \mathbf{P}_{\boldsymbol{\beta}^{*\perp}} (\boldsymbol{X}_i^{\otimes 2} - \mathbf{I}_d) \mathbf{P}_{\boldsymbol{\beta}^{*\perp}} \right\|_{\max} \leq \sqrt{\frac{96\mathbb{E}Y^2 \log d}{cn}},$$

with probability at least $1 - 2d^{-1} - (2^{11}K^4/(\mathbb{E}Y^2)^2 + e)n^{-1}$. Taking into account that $\mathbb{E}Y^2 \leq 4K^2$ we obtain that $C_2 = 384K^2/c$. $\qquad\square$

# E    Proofs for Second Step

**Remark E.1.** For simplicity of presentation we will subtract $n$ from the indexes of the set $S_2$ in the proofs, i.e., instead of having observations indexed in the range $S_2 = \{n+1, \ldots, 2n\}$ we will pretend that our observations are in the range $\{1, \ldots, n\}$.

*Proof of Theorem 3.3.* Take the fixed estimate $\widehat{\mathbf{v}}$ from the first step (recall that $\|\widehat{\mathbf{v}}\|_2 = 1$), and decompose it to:

$$\widehat{\mathbf{v}} = (\widehat{\mathbf{v}}^\top \boldsymbol{\beta}^*)\boldsymbol{\beta}^* + \widehat{\boldsymbol{\beta}}^\perp.$$

By the Pythagorean theorem we have $1 = \|\widehat{\mathbf{v}}\|_2^2 = (\widehat{\mathbf{v}}^\top \boldsymbol{\beta}^*)^2 \|\boldsymbol{\beta}^*\|_2^2 + \|\widehat{\boldsymbol{\beta}}^\perp\|_2^2$, which implies that

$$\|\widehat{\boldsymbol{\beta}}^\perp\|_2 = \sqrt{1 - (\widehat{\mathbf{v}}^\top \boldsymbol{\beta}^*)^2} \leq 1. \tag{E.1}$$

Put $\alpha := c_0 \widehat{\mathbf{v}}^\top \boldsymbol{\beta}^*$ so by Lemma D.2 we have $|\alpha| > \kappa c_0$ with high probability. By formulation (3.6) we have:

$$\frac{1}{2n}\|\mathbf{X}(\widehat{\mathbf{b}} - \alpha\boldsymbol{\beta}^*)\|_2^2 + \lambda_n\|\widehat{\mathbf{b}}\|_1 \leq \frac{1}{n}\left\langle (\boldsymbol{Y} - \overline{\boldsymbol{Y}}) \odot \mathbf{X}\widehat{\mathbf{v}} - \alpha\mathbf{X}\boldsymbol{\beta}^*, \mathbf{X}(\widehat{\mathbf{b}} - \alpha\boldsymbol{\beta}^*) \right\rangle + \nu_n\|\alpha\boldsymbol{\beta}^*\|_1.$$

We handle the empirical process term in Lemma E.3, which also presents the main difficulty in the analysis of the $\ell_1$ regularized least squares procedure. Using this result we conclude that:

$$\frac{1}{2n}\|\mathbf{X}(\widehat{\mathbf{b}} - \alpha\boldsymbol{\beta}^*)\|_2^2 + \nu_n\|\widehat{\mathbf{b}}\|_1 \leq \widetilde{C}\sqrt{\frac{\log d}{n}}\left[\|\widehat{\mathbf{b}} - \alpha\boldsymbol{\beta}^*\|_1 + \frac{1}{\sqrt{n}}\|\mathbf{X}(\widehat{\mathbf{b}} - \alpha\boldsymbol{\beta}^*)\|_2\right] + \nu_n\|\alpha\boldsymbol{\beta}^*\|_1, \tag{E.2}$$

with probability at least $1 - O(n^{-1} + d^{-1})$. We now distinguish two cases. First assume that $\|\mathbf{X}(\widehat{\mathbf{b}} - \alpha\boldsymbol{\beta}^*)\|_2 > 2\widetilde{C}\sqrt{\log d}$. Then (E.2) implies that:

$$\frac{1}{4n}\|\mathbf{X}(\widehat{\mathbf{b}} - \alpha\boldsymbol{\beta}^*)\|_2^2 + \nu_n\|\widehat{\mathbf{b}}\|_1 \leq \widetilde{C}\sqrt{\frac{\log d}{n}}\|\widehat{\mathbf{b}} - \alpha\boldsymbol{\beta}^*\|_1 + \nu_n\|\alpha\boldsymbol{\beta}^*\|_1, \tag{E.3}$$

Next using a standard trick [see, e.g., 4, 6] we have:

$$\|\widehat{\mathbf{b}}\|_1 = \|\widehat{\mathbf{b}}_{S_{\boldsymbol{\beta}^*}}\|_1 + \|\widehat{\mathbf{b}}_{S_{\boldsymbol{\beta}^*}^c}\|_1 \geq \|\alpha\boldsymbol{\beta}^*_{S_{\boldsymbol{\beta}^*}}\|_1 - \|\widehat{\mathbf{b}}_{S_{\boldsymbol{\beta}^*}} - \alpha\boldsymbol{\beta}^*_{S_{\boldsymbol{\beta}^*}}\|_1 + \|\widehat{\mathbf{b}}_{S_{\boldsymbol{\beta}^*}^c}\|_1,$$

$$\|\widehat{\mathbf{b}} - \alpha\boldsymbol{\beta}^*\|_1 = \|\widehat{\mathbf{b}}_{S_{\boldsymbol{\beta}^*}} - \alpha\boldsymbol{\beta}^*_{S_{\boldsymbol{\beta}^*}}\|_1 + \|\widehat{\mathbf{b}}_{S_{\boldsymbol{\beta}^*}^c}\|_1.$$

Selecting $\nu_n \geq 2\widetilde{C}\sqrt{\frac{\log d}{n}}$, the above equalities in combination with (E.3) guarantee that:

$$\frac{1}{4n}\|\mathbf{X}(\widehat{\mathbf{b}} - \alpha\boldsymbol{\beta}^*)\|_2^2 + \nu_n\|\widehat{\mathbf{b}}_{S_{\boldsymbol{\beta}^*}^c} - \alpha\boldsymbol{\beta}^*_{S_{\boldsymbol{\beta}^*}^c}\|_1 \leq 3\nu_n\|\widehat{\mathbf{b}}_{S_{\boldsymbol{\beta}^*}} - \alpha\boldsymbol{\beta}^*_{S_{\boldsymbol{\beta}^*}}\|_1. \tag{E.4}$$

Using Corollary 1 from [29], since clearly $\mathbf{I}_d$ satisfies the RE condition of order $2s$ with constants $(3, 1)$ (i.e., $\forall S \in \binom{[d]}{2s}$ $\forall \boldsymbol{\theta} \in \{\|\boldsymbol{\theta}_{S^c}\|_1 \leq 3\|\boldsymbol{\theta}_S\|_1\}$ we have $\|\boldsymbol{\theta}\|_2 \leq \|\mathbf{I}\boldsymbol{\theta}\|_2$) we can further bound:

$$\frac{1}{4n}\|\mathbf{X}(\widehat{\mathbf{b}} - \alpha\boldsymbol{\beta}^*)\|_2^2 \geq \frac{1}{4 \cdot 8^2}\|\widehat{\mathbf{b}} - \alpha\boldsymbol{\beta}^*\|_2^2, \tag{E.5}$$

with probability at least $1 - c'\exp(-c''n)$ if $n > c'''4^2 s \log d$ where $c', c'', c''' > 0$ are absolute constants. On the above event, (E.4) implies:

$$\frac{1}{4 \cdot 8^2}\|\widehat{\mathbf{b}} - \alpha\boldsymbol{\beta}^*\|_2^2 \leq 3\nu_n\|\widehat{\mathbf{b}}_{S_{\boldsymbol{\beta}^*}} - \alpha\boldsymbol{\beta}^*_{S_{\boldsymbol{\beta}^*}}\|_1 \leq 3\nu_n\sqrt{2s}\|\widehat{\mathbf{b}} - \alpha\boldsymbol{\beta}^*\|_2,$$

where we used Cauchy-Schwartz and the fact that the vector $\widehat{\mathbf{b}}_{S_{\beta^*}} - \alpha\boldsymbol{\beta}^*_{S_{\beta^*}}$ is at most $2s$ sparse. The above inequality gives us that:

$$\|\widehat{\mathbf{b}} - \alpha\boldsymbol{\beta}^*\|_2 \leq 12 \cdot 8^2 \sqrt{2s}\nu_n. \tag{E.6}$$

In the second case when $\|\mathbf{X}(\widehat{\mathbf{b}} - \alpha\boldsymbol{\beta}^*)\|_2 \leq 2\widetilde{C}\sqrt{\log d}$, on the event (E.5) we have:

$$\|\widehat{\mathbf{b}} - \alpha\boldsymbol{\beta}^*\|_2 \leq 32\widetilde{C}\sqrt{\frac{\log d}{n}},$$

and we see that in either case bound (E.6) holds. Before we complete the proof we need the following straightforward result:

**Lemma E.2.** Assume that $n$ is large enough so that $12 \cdot 8^2 \sqrt{2s}\nu_n \leq \kappa c_0/2$. Then with probability at least $1 - O(n^{-1} + d^{-1})$ we have:

$$\min_{\eta \in \{1,-1\}} \left\| \frac{\widehat{\mathbf{b}}}{\|\widehat{\mathbf{b}}\|_2} - \eta\boldsymbol{\beta}^* \right\|_2 \leq \frac{38 \cdot 8^2 \sqrt{2s}\nu_n}{\kappa c_0}$$

Finally notice that $s\sqrt{\frac{\log d}{n}} < R$ implies that $12 \cdot 8^2 \sqrt{2s}\nu_n \leq \kappa c_0/2$ when $R$ is small enough. $\square$

*Proof of Lemma E.2.* Put $r = 12 \cdot 8^2 \sqrt{2s}\nu_n$ for brevity. By (E.6) and the triangle inequality we can conclude that:

$$|\alpha| - r \leq \|\widehat{\mathbf{b}}\|_2 \leq r + |\alpha|.$$

Additionally:

$$\left\| \frac{\widehat{\mathbf{b}}}{\|\widehat{\mathbf{b}}\|_2} - \text{sign}(\alpha)\boldsymbol{\beta}^* \right\|_2 \leq \frac{\|\widehat{\mathbf{b}} - \alpha\boldsymbol{\beta}^*\|_2 + |\|\widehat{\mathbf{b}}\|_2 - |\alpha||}{\|\widehat{\mathbf{b}}\|_2} \leq \frac{2r}{|\alpha| - r} \leq \frac{4r}{|\alpha|} \leq \frac{4r}{\kappa c_0},$$

with the last two inequalities holding with high probability when $r < \kappa c_0/2$ ($\leq |\alpha|/2$ with high probability by Lemma D.2). This completes the proof. $\square$

**Lemma E.3.** There exists a constant $\widetilde{C}$ depending on $f, \varepsilon$ such that:

$$\frac{1}{n}\left\langle (\mathbf{Y} - \overline{\mathbf{Y}}) \odot \mathbf{X}\widehat{\mathbf{v}} - \alpha\mathbf{X}\boldsymbol{\beta}^*, \mathbf{X}(\widehat{\mathbf{b}} - \alpha\boldsymbol{\beta}^*) \right\rangle \leq \widetilde{C}\sqrt{\frac{\log d}{n}}\left[\frac{1}{\sqrt{n}}\|\mathbf{X}(\widehat{\mathbf{b}} - \alpha\boldsymbol{\beta}^*)\|_2 + \|\widehat{\mathbf{b}} - \alpha\boldsymbol{\beta}^*\|_1\right],$$

with probability at least $1 - O(n^{-1} + d^{-1})$.

*Proof of Lemma E.3.* Using Hölder's inequality we obtain:

$$\frac{1}{n}\left\langle (\mathbf{Y} - \overline{\mathbf{Y}}) \odot \mathbf{X}\widehat{\mathbf{v}} - \alpha\mathbf{X}\boldsymbol{\beta}^*, \mathbf{X}(\widehat{\mathbf{b}} - \alpha\boldsymbol{\beta}^*) \right\rangle \leq \frac{1}{n}\|\mathbf{X}^\top[(\mathbf{Y} - \boldsymbol{\mu}) \odot \mathbf{X}\widehat{\mathbf{v}} - \alpha\mathbf{X}\boldsymbol{\beta}^*]\|_\infty\|\widehat{\mathbf{b}} - \alpha\boldsymbol{\beta}^*\|_1 \tag{E.7}$$

$$+ \frac{1}{n}\|(\overline{\mathbf{Y}} - \boldsymbol{\mu}) \odot \mathbf{X}\widehat{\mathbf{v}}\|_2\|\mathbf{X}(\widehat{\mathbf{b}} - \alpha\boldsymbol{\beta}^*)\|_2$$

where we have set $\boldsymbol{\mu} := \mathbb{E}\mathbf{Y}$ for brevity. We first handle the second term. We have $\frac{1}{\sqrt{n}}\|(\overline{\mathbf{Y}} - \boldsymbol{\mu}) \odot \mathbf{X}\widehat{\mathbf{v}}\|_2 = |\overline{Y} - \mu|\frac{1}{\sqrt{n}}\|\mathbf{X}\widehat{\mathbf{v}}\|_2$. Since $Y_i$ is assumed to be sub-exponential by a Bernstein type of inequality we have:

$$\mathbb{P}(|\overline{Y} - \mu| \geq t) \leq 2\exp(-c\min(nt^2/4K^2, nt/2K))$$

where $c$ is an absolute constant. Thus we conclude that $|\overline{Y} - \mu| \leq \frac{2K}{\sqrt{c}}\sqrt{\frac{\log d}{n}}$ with probability at least $2d^{-1}$, for values of $n$ such that $\sqrt{\frac{\log d}{n}} < c$. Also since we have $\mathbf{X}\widehat{\mathbf{v}} \sim \mathcal{N}(0, \mathbf{I}_n)$ we obtain that $\|\mathbf{X}\widehat{\mathbf{v}}\|_2^2 \sim \chi_n^2$. Hence by Chebyshev's inequality we obtain:

$$\mathbb{P}(|\|\mathbf{X}\widehat{\mathbf{v}}\|_2^2/n - 1| \geq t) \leq 2/(nt),$$

and thus by plugging in $t = 1$, we conclude that $\frac{1}{\sqrt{n}}\|\mathbf{X}\widehat{\mathbf{v}}\|_2 \leq \sqrt{2}$ with probability at least $1 - 2n^{-1}$. Hence $\frac{1}{\sqrt{n}}\|(\overline{\mathbf{Y}} - \boldsymbol{\mu}) \odot \mathbf{X}\widehat{\mathbf{v}}\|_2 \leq \widetilde{C}_1\sqrt{\frac{\log d}{n}}$ with probability at least $1 - O(n^{-1}) - 2d^{-1}$.

Next we analyze the sup norm term appearing in inequality (E.7). The first fact we observe is that by construction this term is unbiased since:

$$\mathbb{E}[(Y-\mu)\boldsymbol{X}^{\otimes 2}\boldsymbol{\beta}^*(\widehat{\boldsymbol{v}}^\top\boldsymbol{\beta}^*) - \alpha\boldsymbol{X}^{\otimes 2}\boldsymbol{\beta}^*] + \mathbb{E}[(Y-\mu)\boldsymbol{X}^{\otimes 2}\widehat{\boldsymbol{\beta}}^\perp]$$

$$= \underbrace{\boldsymbol{\beta}^*\mathbb{E}[(Y-\mu)(\boldsymbol{X}^\top\boldsymbol{\beta}^*)^2(\widehat{\boldsymbol{v}}^\top\boldsymbol{\beta}^*) - \alpha\boldsymbol{\beta}^{*\top}\boldsymbol{X}^{\otimes 2}\boldsymbol{\beta}^*]}_{0}$$

$$+ \underbrace{\mathbb{E}[(Y-\mu)\mathbf{P}_{\boldsymbol{\beta}^{*\perp}}\boldsymbol{X}^{\otimes 2}\boldsymbol{\beta}^*(\widehat{\boldsymbol{v}}^\top\boldsymbol{\beta}^*)]}_{0} - \underbrace{\mathbb{E}[\alpha\mathbf{P}_{\boldsymbol{\beta}^{*\perp}}\boldsymbol{X}^{\otimes 2}\boldsymbol{\beta}^*]}_{0}$$

$$+ \underbrace{\mathbb{E}[(Y-\mu)\mathbf{P}_{\boldsymbol{\beta}^{*\perp}}\boldsymbol{X}^{\otimes 2}\widehat{\boldsymbol{\beta}}^\perp]}_{0} + \boldsymbol{\beta}\underbrace{\mathbb{E}[(Y-\mu)\boldsymbol{\beta}^{*\top}\boldsymbol{X}^{\otimes 2}\widehat{\boldsymbol{\beta}}^\perp]}_{0}.$$

Now according to the decomposition in the preceding display, we break down the sup norm term in (E.7) into mean zero terms using the triangle inequality:

$$n^{-1}\|\mathbf{X}^\top[(\boldsymbol{Y}-\boldsymbol{\mu})\odot\mathbf{X}\widehat{\boldsymbol{v}} - \alpha\mathbf{X}\boldsymbol{\beta}^*]\|_\infty \leq n^{-1}\|\mathbf{P}_{\{\boldsymbol{\beta}^*,\widehat{\boldsymbol{\beta}}^\perp\}^\perp}\mathbf{X}^\top[(\boldsymbol{Y}-\boldsymbol{\mu})\odot\mathbf{X}\widehat{\boldsymbol{\beta}}^\perp]\|_\infty \qquad (\text{E.8})$$

$$+ n^{-1}\|\boldsymbol{\beta}^*\boldsymbol{\beta}^{*\top}\mathbf{X}^\top[(\boldsymbol{Y}-\boldsymbol{\mu})\odot\mathbf{X}\widehat{\boldsymbol{\beta}}^\perp]\|_\infty$$

$$+ \frac{n^{-1}}{\|\widehat{\boldsymbol{\beta}}^\perp\|_2^2}\|\widehat{\boldsymbol{\beta}}^\perp(\widehat{\boldsymbol{\beta}}^\perp)^\top\mathbf{X}^\top[(\boldsymbol{Y}-\boldsymbol{\mu})\odot\mathbf{X}\widehat{\boldsymbol{\beta}}^\perp]\|_\infty$$

$$+ n^{-1}\|\boldsymbol{\beta}^*\boldsymbol{\beta}^{*\top}\mathbf{X}^\top[(\boldsymbol{Y}-\boldsymbol{\mu})\odot\mathbf{X}\boldsymbol{\beta}^* - c_0\mathbf{X}\boldsymbol{\beta}^*]\|_\infty$$

$$+ n^{-1}\|\mathbf{P}_{\boldsymbol{\beta}^{*\perp}}\mathbf{X}^\top[(\boldsymbol{Y}-\boldsymbol{\mu})\odot\mathbf{X}\boldsymbol{\beta}^* - c_0\mathbf{X}\boldsymbol{\beta}^*]\|_\infty,$$

where in the last two terms we used the fact that $|\widehat{\boldsymbol{v}}^\top\boldsymbol{\beta}^*| \leq 1$ and $\mathbf{P}_{\{\boldsymbol{\beta}^*,\widehat{\boldsymbol{\beta}}^\perp\}}$ is the projection on the space $\text{span}\{\boldsymbol{\beta}^*,\widehat{\boldsymbol{\beta}}^\perp\}^\perp$. We use Lemma E.4 to control the first term of the decomposition. Lemma E.5 handles the second term, and Lemmas E.7 and E.9 show concentration for the remaining terms. We conclude that there exists a constant $\widetilde{C}_2$ such that:

$$n^{-1}\|\mathbf{X}^\top[(\boldsymbol{Y}-\boldsymbol{\mu})\odot\mathbf{X}\widehat{\boldsymbol{v}} - \alpha\mathbf{X}\boldsymbol{\beta}^*]\|_\infty \leq \widetilde{C}_2\sqrt{\frac{\log d}{n}},$$

with probability at least $1 - O(n^{-1} + d^{-1})$, which is what we aimed to show with $\widetilde{C} = \max(\widetilde{C}_1, \widetilde{C}_2)$.
□

**Lemma E.4.** We have that:

$$\left\|\frac{1}{n}\sum_{i=1}^n(Y_i-\mu)\boldsymbol{X}_i^\top\widehat{\boldsymbol{\beta}}^\perp\boldsymbol{X}_i^\top\mathbf{P}_{\{\boldsymbol{\beta}^*,\widehat{\boldsymbol{\beta}}^\perp\}^\perp}\right\|_\infty \leq \|\widehat{\boldsymbol{\beta}}^\perp\|_2 C_3\sqrt{\frac{\log d}{n}},$$

for an absolute constant $C_3$ depending on $f$ and $\varepsilon$ with probability at least $1 - 2d^{-1} - \frac{\text{Var}(f(Z,\varepsilon) - \mathbb{E}f(Z,\varepsilon))^2}{[\mathbb{E}(f(Z,\varepsilon) - \mathbb{E}f(Z,\varepsilon))^2]^2}n^{-1}$.

*Proof of Lemma E.4.* Notice that $\boldsymbol{X}_i^\top\mathbf{P}_{\{\boldsymbol{\beta}^*,\widehat{\boldsymbol{\beta}}^\perp\}^\perp}$ is independent of $(Y_i-\mu)\boldsymbol{X}_i^\top\widehat{\boldsymbol{\beta}}^\perp$. Hence conditionally on $\boldsymbol{Y}$ and $\mathbf{X}\widehat{\boldsymbol{\beta}}^\perp$ we have

$$\frac{1}{n}\sum_{i=1}^n(Y_i-\mu)\boldsymbol{X}_i^\top\widehat{\boldsymbol{\beta}}^\perp\boldsymbol{X}_i^\top\mathbf{P}_{\{\boldsymbol{\beta}^*,\widehat{\boldsymbol{\beta}}^\perp\}^\perp} \sim \mathcal{N}\Big(0, \frac{1}{n^2}\sum_{i=1}^n(Y_i-\mu)^2(\boldsymbol{X}_i^\top\widehat{\boldsymbol{\beta}}^\perp)^2\mathbf{P}_{\{\boldsymbol{\beta}^*,\widehat{\boldsymbol{\beta}}^\perp\}^\perp}\Big).$$

Next using Chebyshev's inequality we can control the probability of spread about the mean:

$$\mathbb{P}\Big(\Big|\frac{1}{n}\sum_{i=1}^n\frac{(\boldsymbol{X}_i^\top\widehat{\boldsymbol{\beta}}^\perp)^2}{\|\widehat{\boldsymbol{\beta}}^\perp\|_2^2}(Y_i-\mu)^2 - \mathbb{E}(f(Z,\varepsilon) - \mathbb{E}f(Z,\varepsilon))^2\Big| \geq t\Big) \leq \frac{\text{Var}[(f(Z,\varepsilon) - \mathbb{E}f(Z,\varepsilon))^2]}{nt^2},$$

$$(\text{E.9})$$

by setting $t = \mathbb{E}(f(Z,\varepsilon) - \mathbb{E}f(Z,\varepsilon))^2$. Using the fact that $\|\mathbf{P}_{\{\boldsymbol{\beta}^*,\widehat{\boldsymbol{\beta}}^\perp\}^\perp}\|_2 \leq 1$, by a standard normal tail bound and union bound on the event $\frac{1}{n}\sum_{i=1}^n(\boldsymbol{X}_i^\top\widehat{\boldsymbol{\beta}}^\perp)^2(Y_i-\mu)^2 \leq 2\|\widehat{\boldsymbol{\beta}}^\perp\|_2^2\mathbb{E}(f(Z,\varepsilon) - \mathbb{E}f(Z,\varepsilon))^2$ we obtain:

$$\mathbb{P}\Big(\Big\|\frac{1}{n}\sum_{i=1}^n(Y_i-\mu)\boldsymbol{X}_i^\top\widehat{\boldsymbol{\beta}}^\perp\boldsymbol{X}_i^\top\mathbf{P}_{\{\boldsymbol{\beta}^*,\widehat{\boldsymbol{\beta}}^\perp\}^\perp}\Big\|_\infty \geq t\Big) \leq 2d\exp(-nt^2/[4\|\widehat{\boldsymbol{\beta}}^\perp\|_2^2\mathbb{E}(f(Z,\varepsilon) - \mathbb{E}f(Z,\varepsilon))^2]).$$

Select $t = 2\sqrt{2\mathbb{E}(f(Z,\varepsilon) - \mathbb{E}f(Z,\varepsilon))^2}\|\widehat{\boldsymbol{\beta}}^\perp\|_2\sqrt{\frac{\log d}{n}}$ yields the desired bound with

$$C_3 = 2\sqrt{2\mathbb{E}(f(Z,\varepsilon) - \mathbb{E}f(Z,\varepsilon))^2}\|\widehat{\boldsymbol{\beta}}^\perp\|_2.$$

$\square$

**Lemma E.5.** We have that:

$$n^{-1}\|\boldsymbol{\beta}^*\boldsymbol{\beta}^{*\top}\mathbf{X}^\top[(\boldsymbol{Y} - \boldsymbol{\mu}) \odot \mathbf{X}\widehat{\boldsymbol{\beta}}^\perp]\|_\infty \leq C_4\sqrt{\frac{\log d}{n}},$$

for an absolute constant $C_4$ depending on $f$ and $\varepsilon$ with probability at least $1 - 2d^{-1} - \frac{\operatorname{Var} Z^2(f(Z,\varepsilon) - \mathbb{E}f(Z,\varepsilon))^2}{[\mathbb{E}Z^2(f(Z,\varepsilon) - \mathbb{E}f(Z,\varepsilon))^2]^2}n^{-1}$.

*Proof of Lemma E.5.* Notice that $\|\boldsymbol{\beta}^*\|_\infty \leq \|\boldsymbol{\beta}^*\|_2 = 1$ and thus:

$$n^{-1}\|\boldsymbol{\beta}^*\boldsymbol{\beta}^{*\top}\mathbf{X}^\top[(\boldsymbol{Y} - \boldsymbol{\mu}) \odot \mathbf{X}\widehat{\boldsymbol{\beta}}^\perp]\|_\infty \leq n^{-1}|\boldsymbol{\beta}^{*\top}\mathbf{X}^\top[(\boldsymbol{Y} - \boldsymbol{\mu}) \odot \mathbf{X}\widehat{\boldsymbol{\beta}}^\perp]|.$$

Next since $((\widehat{\boldsymbol{\beta}}^\perp)^\top \boldsymbol{X}_i) \perp (\boldsymbol{\beta}^{*\top}\boldsymbol{X}_i)(Y_i - \mu)$, conditioning on $\{(\boldsymbol{\beta}^{*\top}\boldsymbol{X}_i)(Y_i - \mu)\}_{i\in[n]}$ we obtain:

$$\frac{1}{n}\sum_{i=1}^n (\boldsymbol{\beta}^{*\top}\boldsymbol{X}_i)(Y_i - \mu)((\widehat{\boldsymbol{\beta}}^\perp)^\top \boldsymbol{X}_i) \sim \mathcal{N}\Big(0, \frac{\|\widehat{\boldsymbol{\beta}}\|_2^2}{n^2}\sum_{i=1}^n (\boldsymbol{\beta}^{*\top}\boldsymbol{X}_i)^2(Y_i - \mu)^2\Big),$$

Next,

$$\mathbb{P}\Big(\Big|\frac{1}{n}\sum_{i=1}^n (\boldsymbol{\beta}^{*\top}\boldsymbol{X}_i)^2(Y_i - \mu)^2 - \mathbb{E}Z^2(f(Z,\varepsilon) - \mathbb{E}f(Z,\varepsilon))^2\Big| \geq t\Big) \leq \frac{\operatorname{Var}[Z^2(f(Z,\varepsilon) - \mathbb{E}f(Z,\varepsilon))^2]}{nt^2},$$

(E.10)

by setting $t = \mathbb{E}Z^2(f(Z,\varepsilon) - \mathbb{E}f(Z,\varepsilon))^2$ we can control the variance term above. The final bound follows after an application of a standard Gaussian tail bound, where $C_4$ turns out to be $C_4 = \|\widehat{\boldsymbol{\beta}}\|_2 2\sqrt{\mathbb{E}Z^2(f(Z,\varepsilon) - \mathbb{E}f(Z,\varepsilon))^2}$. $\square$

**Lemma E.6.** For large enough values of $n$ we have:

$$\frac{n^{-1}}{\|\widehat{\boldsymbol{\beta}}^\perp\|_2^2}\|\widehat{\boldsymbol{\beta}}^\perp(\widehat{\boldsymbol{\beta}}^\perp)^\top\mathbf{X}^\top[(\boldsymbol{Y} - \boldsymbol{\mu}) \odot \mathbf{X}\widehat{\boldsymbol{\beta}}^\perp]\|_\infty \leq C_5\sqrt{\frac{\log d}{n}},$$

with prob at least $1 - 2d^{-1} - O(n^{-1})$.

*Proof of Lemma E.6.* We have that $\|\widehat{\boldsymbol{\beta}}^\perp\|_\infty \leq \|\widehat{\boldsymbol{\beta}}^\perp\|_2$, and hence:

$$\frac{n^{-1}}{\|\widehat{\boldsymbol{\beta}}^\perp\|_2^2}\|\widehat{\boldsymbol{\beta}}^\perp(\widehat{\boldsymbol{\beta}}^\perp)^\top\mathbf{X}^\top[(\boldsymbol{Y} - \boldsymbol{\mu}) \odot \mathbf{X}\widehat{\boldsymbol{\beta}}^\perp]\|_\infty \leq \frac{n^{-1}}{\|\widehat{\boldsymbol{\beta}}^\perp\|_2}|(\widehat{\boldsymbol{\beta}}^\perp)^\top\mathbf{X}^\top[(\boldsymbol{Y} - \boldsymbol{\mu}) \odot \mathbf{X}\widehat{\boldsymbol{\beta}}^\perp]|.$$

Observe that $\boldsymbol{X}_i^\top\widehat{\boldsymbol{\beta}}^\perp$ is independent from $Y_i - \mu$, and in addition $\boldsymbol{X}_i^\top\widehat{\boldsymbol{\beta}}^\perp \sim \mathcal{N}(0, \|\widehat{\boldsymbol{\beta}}^\perp\|_2^2)$. Hence $(\boldsymbol{X}_i^\top\widehat{\boldsymbol{\beta}}^\perp)^2/\|\widehat{\boldsymbol{\beta}}^\perp\|_2^2 \sim \chi_1^2$. Next we make usage of the decomposition:

$$\frac{1}{n}\sum_i (Y_i - \mu)(\boldsymbol{X}_i^\top\widehat{\boldsymbol{\beta}}^\perp)^2 = \frac{\|\widehat{\boldsymbol{\beta}}^\perp\|_2^2}{n}\sum_i (Y_i - \mu)((\boldsymbol{X}_i^\top\widehat{\boldsymbol{\beta}}^\perp)^2/\|\widehat{\boldsymbol{\beta}}^\perp\|_2^2 - 1) + \frac{\|\widehat{\boldsymbol{\beta}}^\perp\|_2^2}{n}\sum_i (Y_i - \mu).$$

Since $Y_i$ is assumed to be sub-exponential, the second concentrates about zero by Proposition 5.16 in [32]:

$$\mathbb{P}\Big(\Big|n^{-1}\sum_{i=1}^n Y_i - \mu\Big| \geq t\Big) \leq 2\exp(-c\min(nt^2/K^2, nt/K)).$$

Selecting $t = \frac{K}{\sqrt{c}}\sqrt{\frac{\log d}{n}}$ gives a bound on the probability equal to $2d^{-1}$, for values of $n$ large enough so that $\sqrt{\frac{\log d}{n}} \leq \sqrt{c}$. For the remaining term, conditionally on $\{Y_i\}_{i\in[n]}$, by Lemma 1 of [19] we obtain:

$$\mathbb{P}\Big(\Big|n^{-1}\sum_i (Y_i - \mu)((\boldsymbol{X}_i^\top\widehat{\boldsymbol{\beta}}^\perp)^2/\|\widehat{\boldsymbol{\beta}}^\perp\|_2^2 - 1)\Big| \geq 2\sqrt{n^{-1}\sum_{i=1}^n (Y_i - \mu)^2}\sqrt{t} + 2\max_{i\in[n]}|Y_i - \mu|t\Big)$$

$$\leq 2\exp(-nt).$$

(E.11)

Next, by the triangle inequality:

$$\sqrt{n^{-1}\sum_{i=1}^{n}(Y_i-\mu)^2} \le \sqrt{n^{-1}\sum_{i=1}^{n}Y_i^2} + |\mu|, \quad \max_{i\in[n]}|Y_i-\mu| \le \max_{i\in[n]}|Y_i| + |\mu|.$$

The inequalities in the preceding display allow us to reuse the results (D.6) and (D.7) of Lemma D.7. Thus conditioning on these events (E.11) implies:

$$\mathbb{P}\Big(\Big|n^{-1}\sum_{i}(Y_i-\mu)((\boldsymbol{X}_i^\top\widehat{\boldsymbol{\beta}}^\perp)^2/\|\widehat{\boldsymbol{\beta}}^\perp\|_2^2-1)\Big| \ge (2\sqrt{2\mathbb{E}Y^2}+\mu)\sqrt{t}+(4c'K\log n)t\Big) \le 2\exp(-nt),$$

on an event failing with probability at most $\left(\frac{\mathrm{Var}\,Y^2}{[\mathbb{E}Y^2]^2}+e\right)n^{-1}$. Selecting $t=\frac{\log d}{n}$ implies that with probability at least $1-2d^{-1}-\left(\frac{\mathrm{Var}\,Y^2}{[\mathbb{E}Y^2]^2}+e\right)n^{-1}$ we have:

$$\Big|n^{-1}\sum_{i}(Y_i-\mu)((\boldsymbol{X}_i^\top\widehat{\boldsymbol{\beta}}^\perp)^2/\|\widehat{\boldsymbol{\beta}}^\perp\|_2^2-1)\Big| \le [(2\sqrt{2\mathbb{E}Y^2}+\mu)+4c'K]\sqrt{\frac{\log d}{n}},$$

with the probability of failing being at most $\exp(-nt) \le \max(2d^{-1},O(n^{-1}))$. We remind the reader that we are assuming $\log(d)=o(n/\log^2(n))$. This completes the proof with $C_5=\Big((2\sqrt{2\mathbb{E}Y^2}+\mu)+4c'K+\frac{K}{\sqrt{c}}\Big)\|\widehat{\boldsymbol{\beta}}^\perp\|_2$. $\qquad\square$

**Lemma E.7.** We have that:

$$n^{-1}\|\boldsymbol{\beta}^*\boldsymbol{\beta}^{*\top}\mathbf{X}^\top[(\boldsymbol{Y}-\boldsymbol{\mu})\odot\mathbf{X}\boldsymbol{\beta}^*-c_0\mathbf{X}\boldsymbol{\beta}^*]\|_\infty \le C_6\sqrt{\frac{\log d}{n}},$$

with probability at least $1-O(n^{-1})-3d^{-1}$.

*Proof of Lemma E.7.* As in Lemma E.5 we have:

$$n^{-1}\|\boldsymbol{\beta}^*\boldsymbol{\beta}^{*\top}\mathbf{X}^\top[(\boldsymbol{Y}-\boldsymbol{\mu})\odot\mathbf{X}\boldsymbol{\beta}^*-c_0\mathbf{X}\boldsymbol{\beta}^*]\|_\infty \le n^{-1}|\boldsymbol{\beta}^{*\top}\mathbf{X}^\top[(\boldsymbol{Y}-\boldsymbol{\mu})\odot\mathbf{X}\boldsymbol{\beta}^*-c_0\mathbf{X}\boldsymbol{\beta}^*]|,$$

since $\|\boldsymbol{\beta}^*\|_\infty \le 1$. We decompose the right hand side of the preceding display to:

$$\frac{1}{n}\sum_{i=1}^{n}[(\boldsymbol{\beta}^{*\top}\boldsymbol{X}_i)^2Y_i-(c_0+\mu)]+\frac{(c_0+\mu)}{n}\sum_{i=1}^{n}[1-(\boldsymbol{\beta}^{*\top}\boldsymbol{X}_i)^2].$$

To handle the first term one can easily use Chebyshev's inequality to obtain convergence with probability at least $(\log d)^{-1}$. However, to sharpen this rate, we work around the classic Chebyshev's inequality, by making usage of recent concentration results on polynomials of sub-Gaussian random variables proved in [1]. We have the following:

**Lemma E.8.** We have that:

$$\frac{1}{n}\sum_{i=1}^{n}[(\boldsymbol{\beta}^{*\top}\boldsymbol{X}_i)^2Y_i-(c_0+\mu)] \le \widetilde{C}_6\sqrt{\frac{\log d}{n}},$$

with probability at least $1-\max(O(n^{-1}),d^{-1})$.

Usual concentration bounds on the $\chi^2$ distribution can be used to control the second term. Using Lemma 1 of [19] we obtain:

$$\mathbb{P}\Big(\Big|\frac{1}{n}\sum_{i=1}^{n}(\boldsymbol{\beta}^{*\top}\boldsymbol{X}_i)^2-1\Big|\ge 2\sqrt{t}+2t\Big) \le 2\exp(-nt).$$

Select $t=\sqrt{\frac{\log d}{n}}$ to complete the proof assuming that $t<1$ and setting $C_6=4(c_0+\mu)+\widetilde{C}_6$. $\quad\square$

*Proof of Lemma E.8.* First we construct the random variable $Z_i=\eta_i|\boldsymbol{\beta}^{*\top}\boldsymbol{X}_i|^{1/2}|Y_i|^{1/4}$, where $\eta_i$ is a Rademacher random variable. Notice that $Z_i^4=(\boldsymbol{\beta}^{*\top}\boldsymbol{X}_i)^2Y_i$, and hence $\mathbb{E}Z_i^4=(c_0+\mu)$. We now argue that $Z$ is a sub-Gaussian random variable. By Hölder's inequality, and the definition of $\psi_2$ norm we have:

$$\mathbb{E}|Z|^p \le \sqrt{\mathbb{E}|\boldsymbol{\beta}^{*\top}\boldsymbol{X}|^p\mathbb{E}|Y|^{p/2}} \le (p\|\boldsymbol{\beta}^{*\top}\boldsymbol{X}\|_{\psi_2}(\|Y\|_{\psi_1}/2)^{1/2})^{p/2} \le (p(\|Y\|_{\psi_1}/2)^{1/2})^{p/2},$$

where we used that since $\boldsymbol{\beta}^{*\top}\boldsymbol{X} \sim \mathcal{N}(0,1)$ we have $\|\boldsymbol{\beta}^{*\top}\boldsymbol{X}\|_{\psi_2} \leq 1$. Hence $\|Z\|_{\psi_2} \leq (K/2)^{1/4}$, and thus $Z$ is sub-Gaussian as claimed.

For the remaining part recall the notation preceding Theorem B.2. For $f(x) = x^4$ and $F(\boldsymbol{x}) = \sum_{i=1}^n f(x_i)$ we have $\mathbf{D}^\ell F(\boldsymbol{x}) = \mathrm{diag}_d(f^{(\ell)}(x_1), \ldots, f^{(\ell)}(x_n))$ for $\ell \in [4]$. Using the definition of $\psi_2$ norm we can easily estimate $\mathbb{E}[|Z|^\ell] \leq (\sqrt{\ell})^\ell \|Z\|_{\psi_2}^\ell$. To this end we observe the following:

$$\|\mathrm{diag}_\ell\{x_1, \ldots, x_n\}\|_{\mathcal{J}} = \mathbb{1}(\#\mathcal{J} = 1)\|\boldsymbol{x}\|_2 + \mathbb{1}(\#\mathcal{J} \geq 2)\|\boldsymbol{x}\|_{\max}.$$

Hence:

$$\|\mathbb{E}\mathbf{D}^\ell F(\boldsymbol{Z})\|_{\mathcal{J}} \leq [\mathbb{1}(\#\mathcal{J} = 1)\sqrt{n} + \mathbb{1}(\#\mathcal{J} \geq 2)]4!/(4-\ell)!(\sqrt{4-\ell})^{4-\ell}\|Z\|_{\psi_2}^{4-\ell},$$

for $\ell \in [4]$, where with a slight abuse of notation we understand $(\sqrt{4-\ell})^{(4-\ell)} = 1$ when $\ell = 4$. Using the moment estimate of Theorem B.2 we obtain:

$\|F(\boldsymbol{Z}) - \mathbb{E}F(\boldsymbol{Z})\|_k$

$$\leq K_4 \sum_{\ell \in [4]} \|Z\|_{\psi_2}^\ell \sum_{\mathcal{J} \in \mathcal{P}_\ell} k^{\#\mathcal{J}/2}[\mathbb{1}(\#\mathcal{J} = 1)\sqrt{n} + \mathbb{1}(\#\mathcal{J} \geq 2)]\frac{4!(\sqrt{4-\ell})^{4-\ell}\|Z\|_{\psi_2}^{4-\ell}}{(4-\ell)!}$$

$$\leq \widetilde{K}_4[\sqrt{kn} + k^2],$$

where $\mathcal{P}_\ell$ is the set of partitions of $[\ell]$, the absolute constant $K_4$ depends solely on the dimension four, and $\widetilde{K}_4$ on the $\|Z\|_{\psi_2}$ norm and $K_4$. Next by Chebyshev's inequality:

$$\mathbb{P}(n^{-1}|F(\boldsymbol{Z}) - \mathbb{E}F(\boldsymbol{Z})| \geq t) \leq \frac{\widetilde{K}_4^k[\sqrt{k/n} + k^2/n]^k}{t^k}.$$

Applying this inequality with $k = \min(\lceil \log d \rceil, \lceil (n \log d)^{1/4} \rceil)$, and $t = 2e\widetilde{K}_4\sqrt{\frac{\log d}{n}}$ gives us that:

$$\frac{1}{n}\sum_{i=1}^n [(\boldsymbol{\beta}^{*\top}\boldsymbol{X}_i)^2 Y_i - (c_0 + \mu)] \leq \widetilde{C}_5\sqrt{\frac{\log d}{n}},$$

with probability at least $1 - \exp(-\min(\lceil \log d \rceil, \lceil (n \log d)^{1/4} \rceil)) \geq 1 - \max(O(n^{-1}), d^{-1})$ where $\widetilde{C}_5 = 2e\widetilde{K}_4$. This is what we wanted to show. $\qquad\square$

**Lemma E.9.** We have:

$$n^{-1}\|\mathbf{P}_{\boldsymbol{\beta}^*\perp}\mathbf{X}^\top[(\boldsymbol{Y} - \boldsymbol{\mu}) \odot \mathbf{X}\boldsymbol{\beta}^* - c_0\mathbf{X}\boldsymbol{\beta}^*]\|_\infty \leq C_6\sqrt{\frac{\log d}{n}}.$$

with probability at least $1 - O(n^{-1}) - 2d^{-1}$.

*Proof of Lemma E.9.* We have that $\mathbf{P}_{\boldsymbol{\beta}^*\perp}\mathbf{X}^\top$ is independent of $(\boldsymbol{Y} - \boldsymbol{\mu}) \odot \mathbf{X}\boldsymbol{\beta}^* - c_0\mathbf{X}\boldsymbol{\beta}^*$ and thus:

$$\frac{1}{n}\mathbf{P}_{\boldsymbol{\beta}^*\perp}\mathbf{X}^\top[(\boldsymbol{Y} - \boldsymbol{\mu}) \odot \mathbf{X}\boldsymbol{\beta}^* - c_0\mathbf{X}\boldsymbol{\beta}^*] \sim \mathcal{N}\Big(0, \frac{1}{n^2}\sum_{i=1}^n (Y_i - \mu - c_0)^2 (\boldsymbol{\beta}^{*\top}\boldsymbol{X}_i)^2 \mathbf{P}_{\boldsymbol{\beta}^*\perp}\Big).$$

By Chebyshev's inequality we obtain that

$$\left|\frac{1}{n}\sum_{i=1}^n (Y_i - \mu - c_0)^2 (\boldsymbol{\beta}^{*\top}\boldsymbol{X}_i)^2\right| \leq 2\mathbb{E}((f(Z,\varepsilon) - \mu - c_0)^2 Z^2)$$

with probability at least $\frac{\mathrm{Var}((f(Z,\varepsilon) - \mu - c_0)^2 Z^2)}{[\mathbb{E}((f(Z,\varepsilon) - \mu - c_0)^2 Z^2)]^2 n}$. Since $\|\mathbf{P}_{\boldsymbol{\beta}^*\perp}\|_2 \leq 1$, by a standard Gaussian tail bound we obtain:

$$\mathbb{P}\left(\|\mathbf{P}_{\boldsymbol{\beta}^*\perp}\mathbf{X}^\top[(\boldsymbol{Y} - \boldsymbol{\mu}) \odot \mathbf{X}\boldsymbol{\beta}^* - c_0\mathbf{X}\boldsymbol{\beta}^*]\|_\infty \geq t\right) \leq 2d\exp\Big(-\frac{nt^2}{4\mathbb{E}((f(Z,\varepsilon) - \mu - c_0)^2 Z^2)}\Big).$$

Setting $t = 2\sqrt{2\mathbb{E}((f(Z,\varepsilon) - \mu - c_0)^2 Z^2)\frac{\log d}{n}}$, completes the proof. $\qquad\square$