[Reviews · NeurIPS 2016]

Reviewer 1

Summary

This paper establishes a framework for parameter estimation from measurements obtained through an unknown link function. Their framework includes several important special cases such as logistic regression, 1-bit CS and phase retrieval. They propose a convex optimization based method called AGENT to estimate the parameter in their model and prove minimax recovery guarantees demonstrating the price of the unknown link function. The authors use the three applications above as motivating examples throughout.

Qualitative Assessment

The problems this paper addresses are interesting and important, and the authors provide both rigorous theoretical guarantees for their method as well as many numerical experiments. The exposition is clear and motivations are well explained. It would have been nice to see a near-optimal dependence on the number of samples required or at least to see the tradeoff between agnosticity and number of measurements more clearly. However, I still feel the results are strong and place the manuscript in my top 30%. I have only a few minor comments for improvement: 1) Can the authors comment on the role of \eta in the minimax bound, and how one might in practice obtain the best estimate for the actual parameter? 2) It would be nice to see at least one experiment in the main body, or if there isn't room, perhaps a short remark summarizing. 3) In the experiments shown in Figures 3 and 4, can there be a comparison to existing methods?

Confidence in this Review

2-Confident (read it all; understood it all reasonably well)


Reviewer 2

Summary

The paper presents a generalization of ‘phase retrieval’ models that encompasses a wider range of data generated from sparse parameter vectors by using a single index model (SIM) with an unknown link function. These models encompass the traditional phase retrieval as well as the Wirtinger Flow models of data acquisition, with a covariance condition on the observations and the square of the inner product instead of just on the covariance of the observations and the inner product of the predictor vector X and the parameter B. The authors show that several different conditions will lead to the model they describe so that the class of MPR (misspecified phase retrieval) models is ‘sufficiently broad’. They then provide a two-step algorithm to solve these misspecified models – the first step involves solving a regularized semidefinite program that was relaxed from a nonconvex problem which tries to find an estimate for the direction of B (the desired parameter vector) using an empirical surrogate for the mean. They later prove that this step has a supremum which is bounded by the inverse of the size of the problem. The second step is an l1 – regularized refinement step which they show does not increase the computational complexity of the overall problem given few enough repetitions. The authors also have minimax bounds on the class of link functions that they define for additive SIMs. They back this up with a few numerical experiments.

Qualitative Assessment

The acronym AGENT seems like a bit of a stretch to me. I would like to see more numerical experiments as well as comparing to more than just Wirtinger Flow. I know that there are more classical methods for solving problems in classes like these (though perhaps without the same guarantees!), and would maybe like to see it commented on a little more or compared to in your numerical results. The numerical results section feels somewhat threadbare – we do see how the reinforcement strategy is effective, but the lack of comparisons in computational complexity/effort with other established methods makes it hard to see the benefits in computation; the agnosticism to the model is cool but I think it would also be cool to see exactly how AGENT works on the different models in the same plot for comparison. I think the idea is fairly unique – having a general algorithm that provably works with high probability for different models given enough samples is meritorious. If the authors never deal with the complex domain, then classifying the paper as relevant for misspecified phase retrieval seems somewhat like a misnomer to me. Another thing that might be helpful is defining the domain the Y and X reside in, since you have phase retrieval applications, specifying that in certain cases Y is in the real domain vs the complex domain while X has to be in the complex domain (since in the abstract you define B to be wholly real) is fairly important, especially for making comparisons to Wirtinger Flow. I think one thing that would be helpful for the clarity of the reader is to insert reminders for different definitions to avoid frequent backtracking, for example line 167 instead of “satisfy (2.1)”, maybe “satisfy the covariance condition (2.1)”. I don’t quite understand why in line 190 that by making this different observation we just can apply l1 regularized least squares (without an empirical estimate for the mean). Another question I have is regarding equation (3.2), you say that Y~ depends on two directions, but B^ perp is a hyperplane – I think I understand what you mean but this could be slightly confusing? The authors mix notation for the outer product, in line 211 and equation 3.4, the authors use one notation, but in line 221 the authors simply spell it out. I’m not sure how helpful this is for clarity. Notationally sometimes Y bar is used and sometimes mu is used to denote the average. Is one for the empirical average and one for the ‘true’ expectation?

Confidence in this Review

2-Confident (read it all; understood it all reasonably well)


Reviewer 3

Summary

This paper proposes an estimation procedure for misspecified phase retrieval. It consists of solving a cascade of two convex optimization problems, and it has been proved that it is minimax optimal. Thorough numerical results show that the proposed method is effective.

Qualitative Assessment

strong points. -- A novel two-step procedure is proposed. -- The solution generated by the above procedure is proved to be minimax optimal. -- The solution generated by their method is proved to achieve a tight lower bound even when the link function is unknown. weak points. -- I think the author should compare their proposed method with state-of-the-art solver such as nonconvex optimization method [10] in their experiment. In my limited experiences, this type of method seldom suffers from poor local minima. -- The two steps algorithm introduces two tuning parameters lambda_n and nu_n which may not be appealing in practice. -- Based on the result in Lemma 3.1, the optimal solution can be found by solving a sparse PCA problem in Eq (3.4). I think the author should include their empirical comparison with sparse PCA algorithm such as truncated power method (Xiao-Tong Yuan and Tong Zhang, Truncated Power Method for Sparse Eigenvalue Problems).

Confidence in this Review

1-Less confident (might not have understood significant parts)


Reviewer 4

Summary

This paper considers a general framework for tackling phase retrieval problems in the setting where the observations occur via a potentially unknown nonlinear link function. Specifically, they consider the case where one observes $Y = f(X^T \beta^*)$ where $X$ is a Gaussian matrix, $\beta^*$ is an unknown sparse vector, and $f$ is an unknown nonlinearity that satisfies certain properties. The authors consider a general setting which applies to several different potential models (e.g., quadratic observations, magnitude observations, observations with noise both before and/or after the quadratic/magnitude link function, and more. In these settings, the authors provide an analysis of the performance of a simple two-step procedure for estimating $\beta^*$ and also provide minimax lower bounds for this setting.

Qualitative Assessment

This is a well-written paper and seems potentially useful. The plus side is that the framework appears to be quite general, and in any real-world problem one must necessarily deal with uncertainties in the true nature of the observations. On the down side, the analysis seems to be restricted to the Gaussian design matrix setting, which is not particularly practical. I would be inclined to ignore this, but sometimes this plays a huge role and it is at least possible that the algorithm will break down in more realistic settings. Unfortunately, the authors do not provide any simulations (despite promising to do so in the abstract) to verify the practical performance of their algorithm in such a setting. As a less important side note, the acronym for their algorithm is ridiculous and unnecessary. When you are skipping multiple words and grabbing letters from the middle of words to try to make it work, I think you are defeating the purpose of having an acronym. It would be better to just leave the algorithm un-named and let it someone name it after you once it becomes wildly successful!

Confidence in this Review

2-Confident (read it all; understood it all reasonably well)


Reviewer 5

Summary

This paper proposes a model that combines phase estimation with single index models, termed MPR, and an algorithm to estimate the underlying s-sparse parameter vector. The paper is of theoretical nature, and its main contribution is a theoretical characterization of the proposed procedure.

Qualitative Assessment

This is a high quality paper on a very relevant topic. Recent work on high-dimensional SIM estimation can be found in https://arxiv.org/abs/1506.08910

Confidence in this Review

2-Confident (read it all; understood it all reasonably well)